# Introgression of regulatory alleles and a missense coding mutation drive plumage pattern diversity in the rock pigeon

**Anna I Vickrey[1], Rebecca Bruders[1], Zev Kronenberg[2†], Emma Mackey[1‡], Ryan J Bohlender[3], Emily T Maclary[1], Raquel Maynez[1], Edward J Osborne[2], Kevin P Johnson[4], Chad D Huff[3], Mark Yandell[2], Michael D Shapiro[1]***

[1]School of Biological Sciences, University of Utah, Salt Lake City, United States;
[2]Department of Human Genetics, University of Utah, Salt Lake City, United States;
[3]Department of Epidemiology, MD Anderson Cancer Center, University of Texas, Houston, United States; [4]Illinois Natural History Survey, Prairie Research Institute, University of Illinois Urbana-Champaign, Champaign, United States

**\*For correspondence:**
shapiro@biology.utah.edu

**Present address:** †Phase Genomics, Seattle, United States; ‡Department of Molecular and Cell Biology, University of California, Berkeley, Berkeley, United States

**Competing interests:** The authors declare that no competing interests exist.

**Abstract** Birds and other vertebrates display stunning variation in pigmentation patterning, yet the genes controlling this diversity remain largely unknown. Rock pigeons (*Columba livia*) are fundamentally one of four color pattern phenotypes, in decreasing order of melanism: T-check, checker, bar (ancestral), or barless. Using whole-genome scans, we identified *NDP* as a candidate gene for this variation. Allele-specific expression differences in *NDP* indicate *cis*-regulatory divergence between ancestral and melanistic alleles. Sequence comparisons suggest that derived alleles originated in the speckled pigeon (*Columba guinea*), providing a striking example of introgression. In contrast, barless rock pigeons have an increased incidence of vision defects and, like human families with hereditary blindness, carry start-codon mutations in *NDP*. In summary, we find that both coding and regulatory variation in the same gene drives wing pattern diversity, and post-domestication introgression supplied potentially advantageous melanistic alleles to feral populations of this ubiquitous urban bird.
DOI: https://doi.org/10.7554/eLife.34803.001

## Introduction

Vertebrates have evolved a vast array of epidermal colors and color patterns, often in response to natural, sexual, and artificial selection. Numerous studies have identified key genes that determine variation in the types of pigments that are produced by melanocytes (e.g., *Hubbard et al., 2010*; *Manceau et al., 2010*; *Roulin and Ducrest, 2013*; *Domyan et al., 2014*; *Rosenblum et al., 2014*). In contrast, considerably less is known about the genetic mechanisms that determine pigment *patterning* throughout the entire epidermis and within individual epidermal appendages (e.g., feathers, scales, and hairs) (*Kelsh, 2004*; *Protas and Patel, 2008*; *Kelsh et al., 2009*; *Lin et al., 2009*; *Kaelin et al., 2012*; *Lin et al., 2013*; *Eom et al., 2015*; *Poelstra et al., 2015*; *Mallarino et al., 2016*). In birds, color patterns are strikingly diverse among different populations and species, and these traits have profound impacts on mate-choice, crypsis, and communication (*Hill and McGraw, 2006*).

The domestic rock pigeon (*Columba livia*) displays enormous phenotypic diversity among over 350 breeds, including a wide variety of plumage pigmentation patterns that also vary within breeds (*Shapiro and Domyan, 2013*; *Domyan and Shapiro, 2017*). Some of these pattern phenotypes are found in feral and wild populations as well (*Johnston and Janiga, 1995*). A large number of genetic loci contribute to pattern variation in rock pigeons, including genes that contribute in an additive

**eLife digest** The rock pigeon is a familiar sight in urban settings all over the world. Domesticated thousands of years ago and still raised by hobbyists, there are now more than 350 breeds of pigeon. These breeds have a spectacular variation in anatomy, feather color and behavior. Color patterns are important for birds in species recognition, mate choice and camouflage. Pigeon fanciers have long observed that color patterns can be linked to health problems, such as lighter birds suffering more often from poor vision.

In addition, pigeons with certain pigment patterns are more likely to survive and reproduce in urban habitats. But despite centuries of pigeon-breeding and the abundance of rock pigeons in urban spaces, how pigeons generate such different feather color patterns, is still largely a mystery.

Vickrey et al. sequenced the genomes of pigeons with different patterns and found that a gene called *NDP* played an important role in wing pigmentation. In birds with darker patterns (called checker and T-check) the gene *NDP* was expressed more in their feathers, but the gene itself was not altered. The lightest colored birds (barless patterned), however, had a mutation in the *NDP* gene itself that led to less pigmentation.

The NDP mutation found in barless pigeons is the same as one that is sometimes found in the human version of NDP, where it is linked to hereditary blindness. Vickrey et al. also discovered that the darker patterns most likely arose from breeding of the rock pigeon with a different species, the African speckled pigeon, something pigeon fanciers have suspected for some time.

The findings could help to parse out the different functions of the *NDP* gene in both pigeons and humans. Mutations in the *NDP* gene in humans typically cause a range of neurological problems in addition to loss of sight, but in barless pigeons, the mutation appears to cause only vision defects. These findings suggest that a specific part of the gene is particularly important for vision in birds and humans, and shed light on the surprisingly complex evolutionary history of the rock pigeon.
DOI: https://doi.org/10.7554/eLife.34803.002

fashion and others that epistatically mask the effects of other loci (*Van Hoosen Jones, 1922*; *Hollander, 1937*; *Sell, 2012*; *Domyan et al., 2014*). Despite the genetic complexity of the full spectrum of plumage pattern diversity in pigeons, classical genetic experiments demonstrate that major wing shield pigmentation phenotypes are determined by an allelic series at a single locus (*C*, for 'checker' pattern) that produces four phenotypes: T-check ($C^T$ allele, also called T-pattern), checker (*C*), bar (+), and barless (*c*), in decreasing order of dominance and melanism (*Figure 1A*) (*Bonhote and Smalley, 1911*; *Hollander, 1938a, 1983b*; *Levi, 1986*; *Sell, 2012*). Bar is the ancestral phenotype (*Darwin, 1859*; *Darwin, 1868*), yet checker and T-check can occur at higher frequencies than bar in urban feral populations, suggesting a fitness advantage in areas of dense human habitation (*Goodwin, 1952*; *Obukhova and Kreslavskii, 1984*; *Johnston and Janiga, 1995*; *Čanády and Mošanský, 2013*).

Color pattern variation is associated with several important life history traits in feral pigeon populations. For example, checker and T-check birds have higher frequencies of successful fledging from the nest, longer (up to year-round) breeding seasons, and can sequester more toxic heavy metals in plumage pigments through chelation (*Petersen and Williamson, 1949*; *Lofts et al., 1966*; *Murton et al., 1973*; *Janiga, 1991*; *Chatelain et al., 2014*; *2016*). Relative to bar, checker and T-check birds also have reduced fat storage and, perhaps as a consequence, lower overwinter adult survival rates in harsh rural environments (*Petersen and Williamson, 1949a*; *Jacquin et al., 2012*). Female pigeons prefer checker mates to bars, so sexual selection probably influences the frequencies of wing pigmentation patterns in feral populations as well (*Burley, 1977*; *1981*; *Johnston and Johnson, 1989*). In contrast, barless, the recessive and least melanistic phenotype, is rarely observed in feral pigeons (*Johnston and Janiga, 1995*). In domestic populations, barless birds have a higher frequency of vision defects, sometimes referred to as 'foggy' vision (*Hollander and Miller, 1981*; *Hollander, 1983b*; *Mangile, 1987*), which could negatively impact fitness in the wild.

In this study, we investigate the molecular basis and evolutionary history underlying wing pattern diversity in pigeons. We discover both coding and regulatory variation at a single candidate gene,

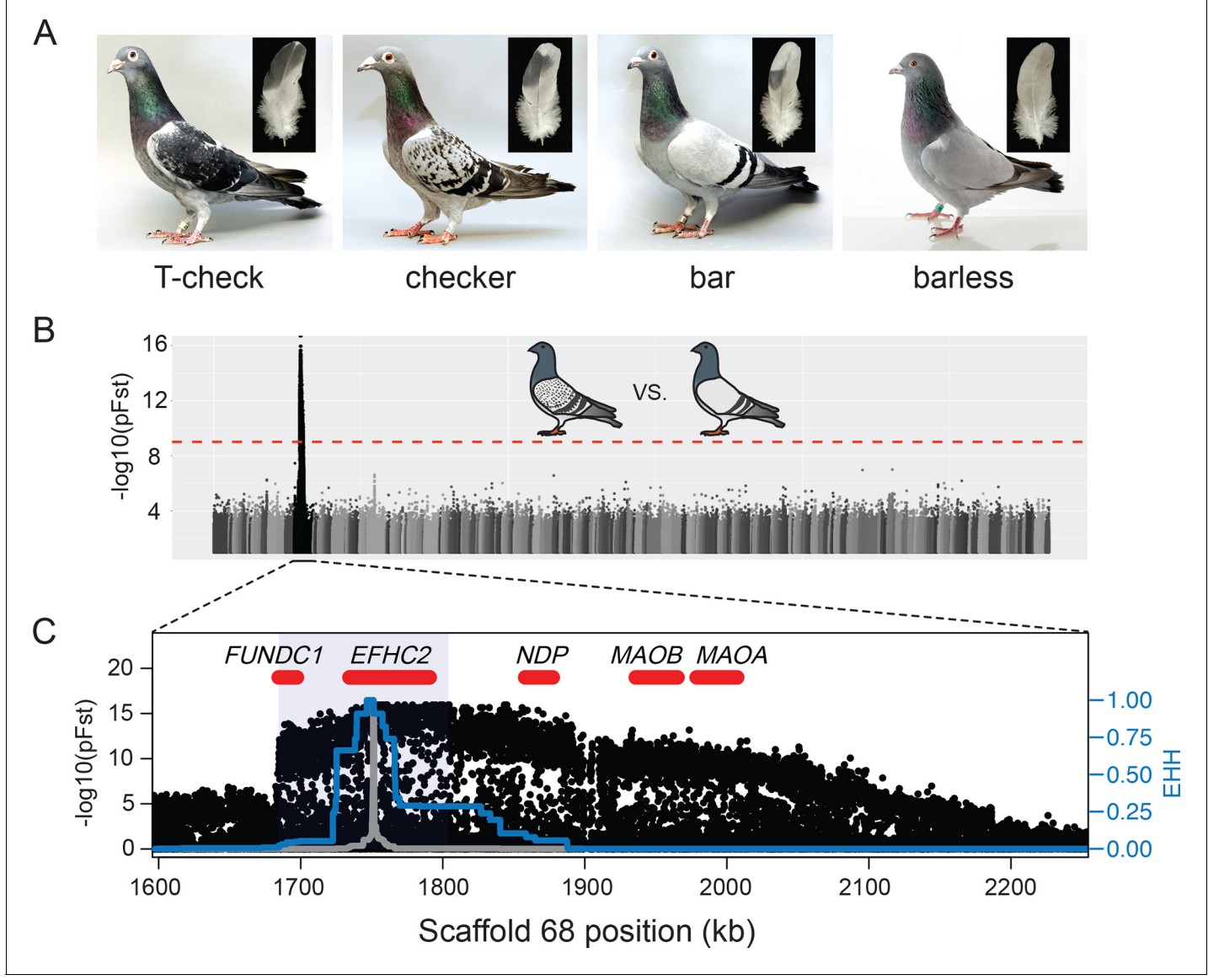

**Figure 1.** A single genomic region is associated with rock pigeon (*C. livia*) wing pigmentation pattern. (**A**) Four classical wing pattern pigmentation phenotypes, shown in decreasing order of genetic dominance and melanism (left to right): T-check, checker, bar, and barless. Photos courtesy of the Genetics Science Learning Center (http://learn.genetics.utah.edu/content/pigeons). (**B**) Whole-genome pFst comparisons between the genomes of bar (n = 17) and checker (n = 24) pigeons. Dashed red line marks the genome-wide significance threshold (9.72e-10). (**C**) Detail of pFst peak shows region of high differentiation on Scaffold 68. Five genes within the region are shown in red. Blue shading marks the location of the smallest shared haplotype common to all checker and T-check birds. Haplotype homozygosity in the candidate region extends further for checker and T-check birds (blue trace) than for bar birds (gray), a signature of positive selection for the derived alleles. Extended haplotype homozygosity (EHH) was measured from focal position 1,751,072 following the method of *Sabeti et al. (2007)*.

DOI: https://doi.org/10.7554/eLife.34803.003

The following figure supplements are available for figure 1:

**Figure supplement 1.** Variation in wing shield color pattern among pigeons with checker alleles in the Scaffold 68 candidate region.
DOI: https://doi.org/10.7554/eLife.34803.004

**Figure supplement 2.** Whole genome pFst comparisons to identify a candidate genomic region differentiated between birds with different wing pattern phenotypes.
DOI: https://doi.org/10.7554/eLife.34803.005

**Figure supplement 3.** EFHC2 amino acid sequences of pigeons and other amniotes (residues 525–604).
DOI: https://doi.org/10.7554/eLife.34803.006

**Figure supplement 4.** *C* locus genotypes segregate with phenotype in an $F_2$ intercross.

*Figure 1 continued*

DOI: https://doi.org/10.7554/eLife.34803.007

and a polymorphism linked with pattern variation within and between species that likely resulted from interspecies hybridization.

## Results and discussion

### A genomic region on Scaffold 68 is associated with wing pattern phenotype

To identify the genomic region containing the major wing pigmentation pattern locus, we used a probabilistic measure of allele frequency differentiation (pFst; *Domyan et al., 2016*) to compare the resequenced genomes of bar pigeons to genomes of pigeons with either checker or T-check patterns (*Figure 1A*). Checker and T-check birds were grouped together because these two patterns are sometimes difficult to distinguish, even for experienced hobbyists. Checker birds are typically less pigmented than T-check birds, but genetic modifiers of pattern phenotypes can minimize this difference (see *Figure 1—figure supplement 1* for examples of variation). A two-step whole-genome scan (see Materials and methods; *Figure 1B and C*, *Figure 1—figure supplement 2*) identified a single ~103 kb significantly differentiated region on Scaffold 68 that was shared by all checker and T-check birds (position 1,702,691–1,805,600 of the Cliv_1.0 pigeon genome assembly, *Shapiro et al. (2013)*; p=1.11e-16, genome-wide significance threshold = 9.72e-10). The minimal shared region was defined by haplotype breakpoints in a homozygous checker and a homozygous bar bird, and is highly differentiated from the same region in bar (63.28% mean sequence similarity at informative sites). This region is hereafter referred to as the minimal checker haplotype.

As expected for the well-characterized allelic series at the *C* locus, we also found that a broadly overlapping region of Scaffold 68 was highly differentiated between the genomes of bar and barless birds (p=3.11e-15, genome-wide significance threshold = 9.71e-10; *Figure 1—figure supplement 2*). Together, these whole-genome comparisons identified a single genomic region corresponding to the wing-pattern *C* locus.

### A copy number variant is associated with variation in melanistic wing patterns

To identify genetic variants associated with the derived checker and T-check phenotypes, we first compared annotated protein-coding genes throughout the genome. We found a single, predicted, fixed change in EFHC2 (Y572C, *Figure 1—figure supplement 3*) in checker and T-check birds relative to bar birds (VAAST; *Yandell et al., 2011*). However, this same amino acid substitution is also found in *Columba rupestris*, a closely related species to *C. livia* that has a bar wing pattern. Thus, the Y572C substitution is not likely to be causative for the checker or T-check pattern, nor is it likely to have a strong impact on protein function (MutPred2 score 0.468, no recognized affected domain; PolyPhen-2 score 0.036, benign; *Adzhubei et al., 2010*; *Pejaver et al., 2017*).

Next, we examined sequence coverage across the checker haplotype and discovered a copy number variable (CNV) region (approximate breakpoints at Scaffold 68 positions 1,790,000 and 1,805,600). Based on normalized read-depths of resequenced birds, we determined that the CNV region has one, two, or four copies per chromosome. Bar birds (n = 12) in our resequencing panel always had a total of two copies in the CNV region (one on each chromosome), but most checker (n = 5 of 7) and T-check (n = 2 of 2) genomes examined had additional copies of the CNV (*Figure 2A*). Using a PCR assay to amplify across the breakpoints in birds with more than one copy per chromosome, we determined that additional copies result from tandem repeats. We found no evidence that the checker haplotype contains an inversion based on mapping of paired-end reads at the CNV breakpoints (WHAM; *Kronenberg et al., 2015*). In addition, we were able to amplify unique PCR products that span the outer CNV breakpoints (data not shown), suggesting that there are no inversions within the CNV region.

Consistent with the dominant inheritance pattern of the phenotype, all checker and T-check birds had at least one copy of the checker haplotype. However, the fact that some checker birds had only

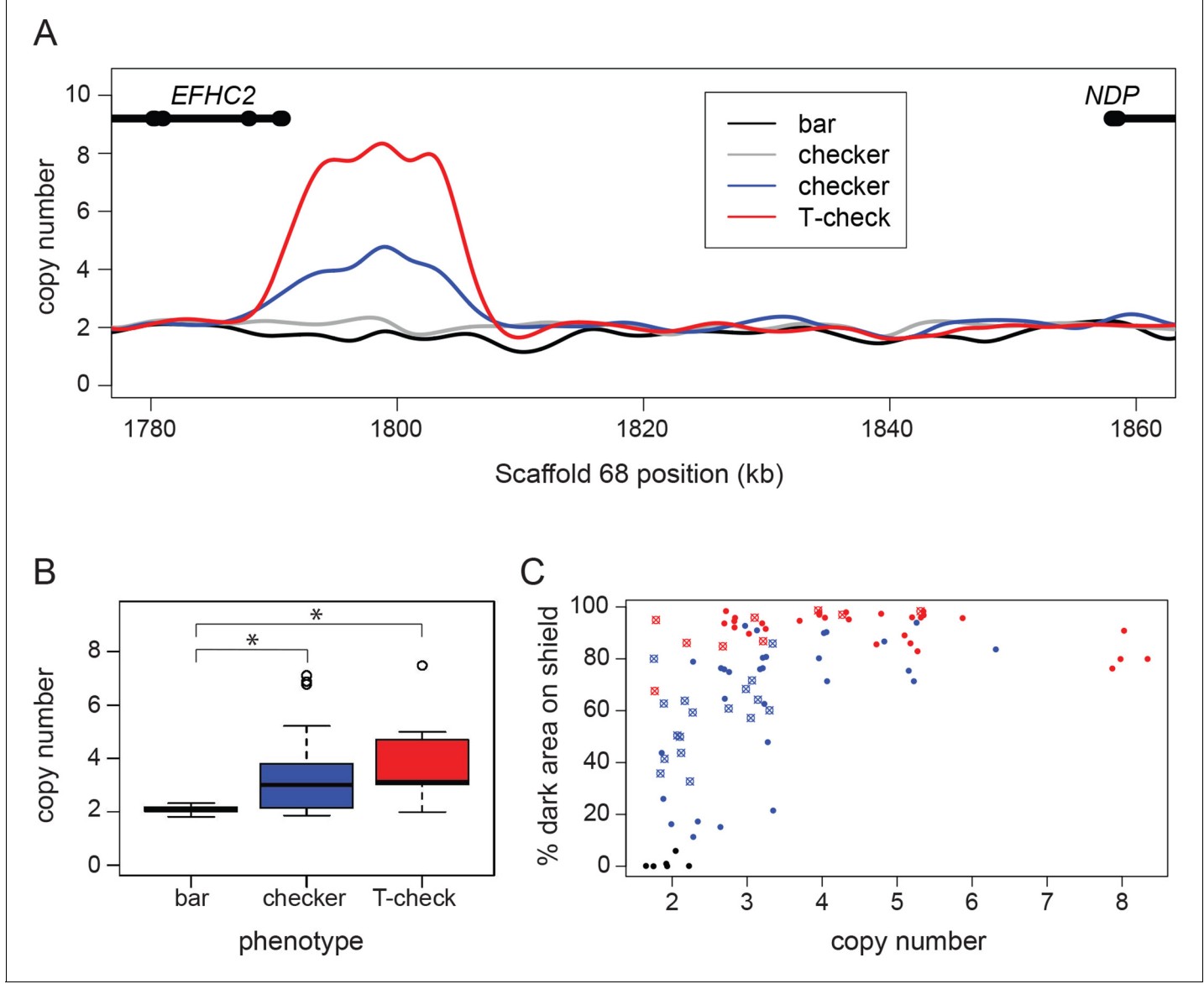

**Figure 2.** A copy number variant (CNV) in the candidate region is associated with T-check and checker phenotypes. (A) Normalized read depths from resequenced birds are plotted in the candidate region between *EFHC2* and *NDP* on Scaffold 68. Thickened portions of gene models represent exons and thin portions are introns. Representative individual read depth traces are shown for the following: black for bar *C. livia*, grey for checker *C. livia* individuals without additional copies of the CNV, blue for checker *C. livia* individuals with additional copies of the CNV region, red for T-check *C. livia*. (B) CNV quantification for 94 birds (20 bar, 56 checker, and 18 T-check). Checker and T-check phenotypes (as reported by breeders) were associated with increased copy numbers (p=2.1e-05). (C) CNV and phenotype quantification for an additional 84 birds, including 26 feral pigeons. Increased copy number was associated with an increase in dark area on the wing shield ($r^2$ = 0.46, linear regression). Points are colored by reported phenotype and origin: bar, black; checker, blue; T-check, red; domestic breeds, filled circle points; ferals, cross points.
DOI: https://doi.org/10.7554/eLife.34803.008

The following source data and figure supplement are available for figure 2:

**Source data 1.** Taqman copy number assay results represented in *Figure 2B*.
DOI: https://doi.org/10.7554/eLife.34803.010

**Source data 2.** Taqman copy number assay and phenotype quantification results represented in *Figure 2C*.
DOI: https://doi.org/10.7554/eLife.34803.011

**Figure supplement 1.** CNV is associated with darker wing shield pigmentation.
DOI: https://doi.org/10.7554/eLife.34803.009

one copy of the CNV region on each chromosome demonstrates that a copy number increase is not necessary to produce melanistic phenotypes. Pedigree analysis of a laboratory cross also confirmed perfect co-segregation of the checker haplotype and phenotype (*Figure 1—figure supplement 4*, *Supplementary file 1*). Thus, a checker haplotype on at least one chromosome appears to be necessary for the dominant melanistic phenotypes, but additional copies of the CNV region are not.

In a larger sample of pigeons, we found a significant association between copy number and phenotype (TaqMan assay; pairwise Wilcoxon test, p=2.1e-05). Checker (n = 40 of 55) and T-check (n = 15 of 18) patterns are usually associated with expansion of the CNV, but pigeons with the bar pattern (n = 20) never had more than two copies in total (one copy on each chromosome; *Figure 2B*). Although additional copies of the CNV only occurred in checker and T-check birds, we did not observe a consistent number of copies associated with either phenotype. This could be due to a variety of factors, including modifiers that darken genotypically-checker birds to closely resemble T-check (*Van Hoosen Jones, 1922*; *Sell, 2012*) and environmental factors such as temperature-dependent darkening of the wing shield during feather development (*Podhradsky, 1968*).

Due to the potential ambiguity in categorical phenotyping, we next measured the percent of pigmented area on the wing shield and tested for associations between copy number and the percentage of pigmented wing-shield area. We phenotyped and genotyped an additional 63 birds from diverse domestic breeds as well as 26 feral birds, and found that estimated copy number in the variable region was correlated with the amount of dark pigment on the wing shield (nonlinear least squares regression, followed by $r^2$ calculation; $r^2 = 0.46$) (*Figure 2C*). This correlation was a better fit to the regression when ferals were excluded ($r^2 = 0.68$, *Figure 2—figure supplement 1*), possibly because numerous pigmentation modifiers (e.g., *sooty* and *dirty*) are segregating in feral populations (*Hollander, 1938a*; *Johnston and Janiga, 1995*). Together, our analyses show that the minimal checker haplotype is associated with increased pigmentation on the wing shield plumage, resulting in qualitative variation between bar and checker (including T-check) phenotypes. Furthermore, copy number variation is found only in checker haplotypes, and higher numbers of copies are associated with quantitative pigmentation increases in checker and T-check birds only.

## *NDP* is differentially expressed in feather buds of different wing pattern phenotypes

The CNV that is associated with wing pattern variation resides between two genes, *EFHC2* and *NDP*. *EFHC2* is a component of motile cilia, and mouse mutants have juvenile myoclonic epilepsy (*Linck et al., 2014*). In humans, allelic variation in *EFHC2* is also associated with differential fear responses and social cognition (*Weiss et al., 2007*; *Blaya et al., 2009*; *Startin et al., 2015*; but see *Zinn et al., 2008*). However, *EFHC2* has not been implicated in pigmentation phenotypes in any organism. *NDP* encodes a secreted ligand that activates *WNT* signaling by binding to its only known receptor FZD4 and its co-receptor LRP5 (*Smallwood et al., 2007*; *Hendrickx and Leyns, 2008*; *Deng et al., 2013*; *Ke et al., 2013*). Notably, *NDP* is one of many differentially expressed genes in the feathers of closely related crow subspecies that differ, in part, by the intensity of plumage pigmentation (*Poelstra et al., 2015*). Furthermore, FZD4 is a known melanocyte stem cell marker (*Yamada et al., 2010*). Thus, based on expression variation in different crow plumage phenotypes, and the expression of its receptor in pigment cell precursors, *NDP* is a strong candidate for pigment variation in pigeons. NDP is a short-range signal (*Niehrs, 2004*), so we suspect that this ligand is secreted by melanocytes themselves or by cells in close proximity to them.

The CNV in the intergenic space between *EFHC2* and *NDP* in the candidate region, coupled with the lack of candidate coding variants between bar and checker haplotypes, led us to hypothesize that the CNV region might contain regulatory variation that could alter expression of one or both neighboring genes. To test this possibility, we performed qRT-PCR on RNA harvested from regenerating wing shield feathers of bar, checker, and T-check birds. *EFHC2* was not differentially expressed between bar and either checker or T-check patterned feathers (p=0.19, pairwise Wilcoxon test, p-value adjustment method: fdr), although expression levels differed slightly between the checker and T-check patterned feathers (p=0.046, *Figure 3A*). Expression levels of other genes adjacent to the minimal checker haplotype region also did not vary by phenotype (*Figure 3—figure supplement 1*).

In contrast, expression of *NDP* was significantly increased in checker feathers – and even higher in T-check feathers – relative to bar feathers (*Figure 3A*) (bar-checker comparison, p=1.9e-05; bar-T-

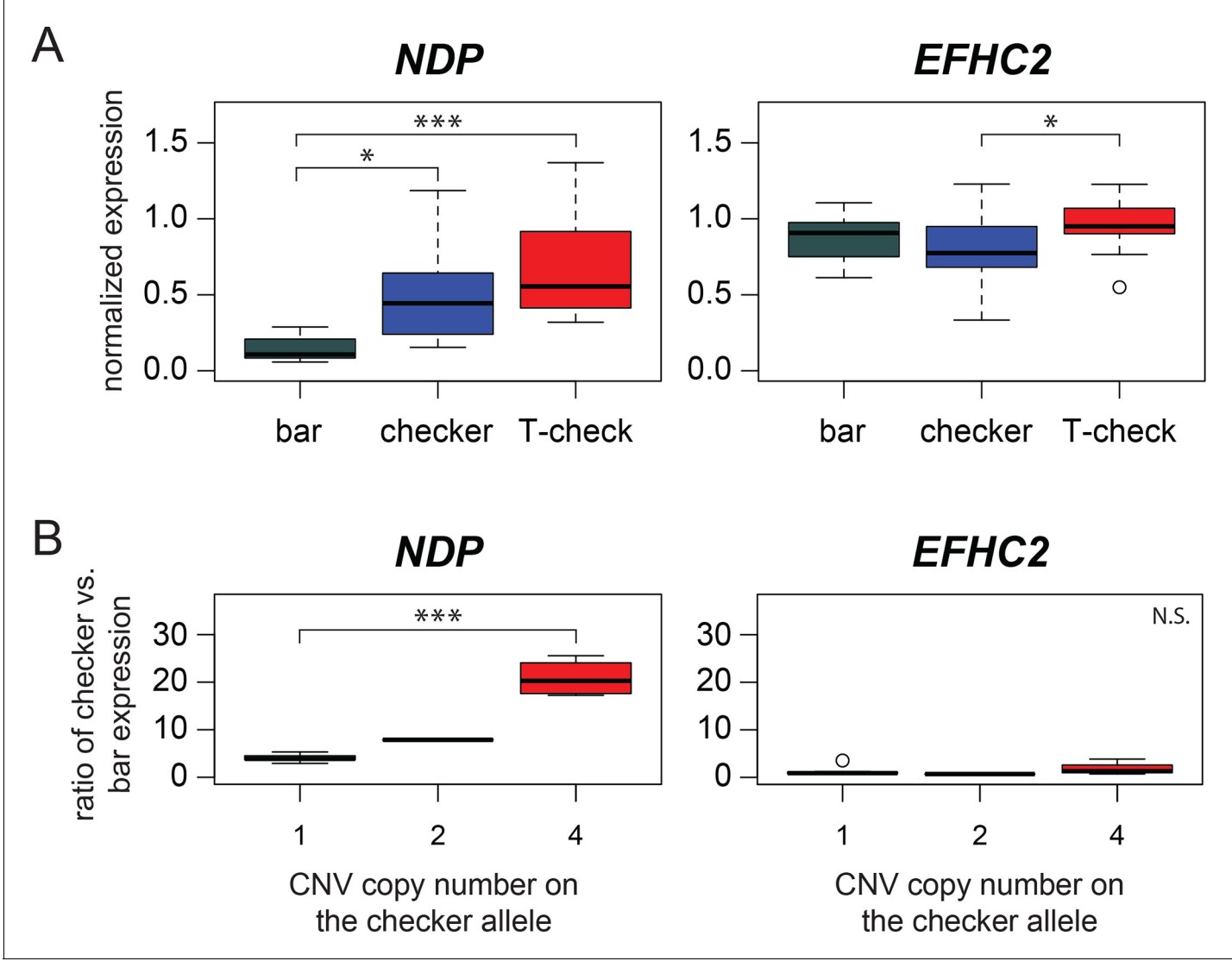

**Figure 3.** Expression differences in *NDP*, but not *EFHC2*, indicate *cis*-regulatory differences associated with pigmentation phenotypes. (**A**) qRT-PCR assays demonstrate higher expression of *NDP* in regenerating feathers of checker and T-check birds than in bar birds. Expression levels of *EFHC2* are indistinguishable between bar and melanistic phenotypes (p=0.19), although checker and T-check differed from each other (p=0.046). (**B**) Allele-specific expression assay in regenerating feathers from heterozygous bar/checker birds for *NDP* and *EFHC2*. Copies of the CNV region on the checker chromosome were quantified using a custom Taqman assay. Boxes span the first to third quartiles, bars extend to minimum and maximum observed values, black line indicates median. Expression of *EFHC2* alleles were not significantly different, and checker alleles of *NDP* showed higher expression than the bar allele; p=0.0028 for two-sample t-test between 1 vs. 4 copies, p=1.84e-06 for glm regression.

DOI: https://doi.org/10.7554/eLife.34803.012

The following source data and figure supplements are available for figure 3:

**Source data 1.** qRT-PCR source data represented in *Figure 3A*, and *Figure 3—figure supplement 1*.

DOI: https://doi.org/10.7554/eLife.34803.016

**Source data 2.** Allele-specific expression assays source data represented in *Figure 3B* and *Figure 3—figure supplement 3*.

DOI: https://doi.org/10.7554/eLife.34803.017

**Figure supplement 1.** Expression of genes involved in pigmentation and genes in the candidate region.

DOI: https://doi.org/10.7554/eLife.34803.013

**Figure supplement 2.** *NDP* expression varies by copy number and phenotype.

DOI: https://doi.org/10.7554/eLife.34803.014

**Figure supplement 3.** Allele-specific expression assay for *NDP* in regenerating feathers from dorsal body and tail feathers.

DOI: https://doi.org/10.7554/eLife.34803.015

check, p=1.0e-08; checker-T-check, p=0.0071; pairwise Wilcoxon test, all comparisons were significant at a false discovery rate of 0.05). Moreover, when qRT-PCR expression data for checker and T-check feathers were grouped by copy number instead of categorical phenotype, the number of CNV copies was positively associated with *NDP* expression level (*Figure 3—figure supplement 2*). Thus, expression of *NDP* is positively associated with both increased melanism (categorical pigment pattern phenotype) and CNV genotype.

The increase in *NDP* expression could be the outcome of at least two molecular mechanisms. First, one or more regulatory elements in the CNV region (or elsewhere on the same DNA strand) could increase expression of *NDP* in *cis*. Such changes would only affect expression of the allele on the same chromosome (*Wittkopp et al., 2004*). Second, *trans*-acting factors encoded within the minimal checker haplotype (e.g., *EFHC2* or an unannotated feature) could increase *NDP* expression, resulting in an upregulation of *NDP* alleles on both chromosomes.

To distinguish between these possibilities, we carried out allele-specific expression assays (*Domyan et al., 2014*; *2016*) on the regenerating wing shield feathers of birds that were heterozygous for bar and checker alleles in the candidate region (checker alleles with one, two, or four copies of the CNV). In the common *trans*-acting cellular environment of heterozygous birds, checker alleles of *NDP* were more highly expressed than bar alleles, and these differences were further amplified in checker alleles with more copies of the CNV (*Figure 3B*) (p=0.0028 for two-sample t-test between 1 vs. 4 copies, p=1.84e-06 for generalized linear model regression; ratios of checker:bar expression for 1- and 4-copy checker alleles were significantly different than 1:1, p≤0.002 for each comparison). In comparison, transcripts of *EFHC2* from checker and bar alleles were not differentially expressed in the heterozygote background (*Figure 3B*) (p=0.55 for two-sample t-test between 1 vs. 4 copies, p=0.47 for linear regression; ratios of checker:bar expression for 1- and 4-copy checker alleles were not significantly different than 1:1, p>0.3 for each comparison). Checker alleles of *NDP* were also more highly expressed in feathers from other body regions (tail and dorsum, *Figure 3—figure supplement 3*), even though the pigment pattern on these regions is generally similar in bar and checker birds (e.g., both phenotypes have a dark band on the tail). Together, our expression studies indicate that a *cis*-acting regulatory change drives increased expression of *NDP* in pigeons with more melanistic plumage patterns, but does not alter expression of *EFHC2* or other nearby genes. Furthermore, because *NDP* expression increases with additional copies of the CNV, the regulatory element probably resides within the CNV itself.

To search for known enhancers in the CNV region, we mapped elements from the VISTA (*Visel et al., 2007*) and REPTILE (*He et al., 2017*) enhancer datasets to the pigeon genome. We found no hits within the minimal haplotype from the VISTA dataset and 12 hits from the REPTILE dataset (*Supplementary file 2*). Of these 12, one hit was within the CNV region (Scaffold 68: 1,795,453–1,795,511). However, this lone mouse enhancer (ENSMUSR00000084784, http://uswest.ensembl.org/Mus_musculus/) is not known to regulate *EFHC2* or *NDP* in mice, and is located on a mouse chromosome that is not orthologous to pigeon Scaffold 68. Further functional work will be required to assess whether this or other sequences in the CNV region act as regulatory elements in *C. livia*.

## A missense mutation at the start codon of *NDP* is associated with barless

In humans, mutations in *NDP* can result in Norrie disease, a recessively-inherited disorder characterized by a suite of symptoms including vision deficiencies, intellectual and motor impairments, and auditory deficiencies (*Norrie, 1927*; *Warburg, 1961*; *Holmes, 1971*; *Chen et al., 1992*; *Sims et al., 1992*). Protein-coding mutations in *NDP*, including identical mutations segregating within single-family pedigrees, result in variable phenotypic outcomes, including incomplete penetrance (*Meindl et al., 1995*; *Berger, 1998*; *Allen et al., 2006*). Intriguingly, barless pigeons also have an increased incidence of vision deficiencies and, as in humans with certain mutant alleles of *NDP*, this phenotype is not completely penetrant (*Hollander, 1983b*). Thus, based on the known allelism at the *C* locus, the nomination of regulatory changes at *NDP* as candidates for the *C* and $C^T$ alleles, and the vision-related symptoms of Norrie disease, *NDP* is also a strong candidate for the barless phenotype (*c* allele).

To test this prediction, we used VAAST to scan the resequenced genomes of 9 barless pigeons and found that all were homozygous for a nonsynonymous protein-coding change at the start codon

of *NDP* that was perfectly associated with the barless wing pattern phenotype (*Figure 4*, *Figure 1—figure supplement 2*). We detected no other genes with fixed coding changes or regions of significant allele frequency differentiation (pFst) elsewhere in the genome. We genotyped an additional 14 barless birds and found that all were homozygous for the same start-codon mutation. The barless mutation is predicted to truncate the amino terminus of the NDP protein by 11 amino acids, thereby disrupting the 24-amino acid signal peptide sequence (www.uniprot.org, Q00604 NDP_Human). *NDP* is still transcribed and detectable by RT-PCR in regenerating barless feathers; therefore, we speculate that the start-codon mutation might alter the normal secretion of the protein into the extracellular matrix (*Gierasch, 1989*).

In humans, coding mutations in *NDP* are frequently associated with a suite of neurological deficits. In pigeons, however, only wing pigment depletion and vision defects are reported in barless homozygotes. Remarkably, two human families segregating Norrie disease have only vision defects, and like barless pigeons, these individuals have start-codon mutations in *NDP* (*Figure 4*) (*Isashiki et al., 1995*). Therefore, signal peptide mutations might affect a specific subset of developmental processes regulated by *NDP*, while leaving other (largely neurological) functions intact. *NDP* is critical for retinal vascular formation (*Xu et al., 2004*) and hedgehog-dependent retinal progenitor proliferation (*McNeill et al., 2013*) in mammals, and we speculate that one or both of these processes is affected by the start codon mutations in pigeons as well. In summary, wing pattern phenotypes in pigeons are associated with the evolution of both regulatory (checker, T-check) and coding (barless) changes in the same gene, and barless pigeons share a partially-penetrant visual deficiency with human patients who have start-codon substitutions.

Future work will test whether the barless (and human) start-codon mutations affect extracellular secretion of NDP, and how *NDP* expression directly or indirectly regulates melanocyte activity. Sharp boundaries define the heavily pigmented areas of checker feathers (*Figure 1*, *Figure 1—figure supplement 1*), similar to intra-feather patterns in other species that are mediated by both activity of melanocytes and the topological distribution of their progenitors (*Lin et al., 2013*; *Chen et al., 2015*). Considerably more is known about the molecular control of plumage structure and color than pigmentation pattern, based in part on experiments to manipulate gene expression in vivo by viral infection and in explants by protein misexpression (*Harris et al., 2002*; *Yu et al., 2002*; *Harris et al., 2005*; *Chen et al., 2015*; *Boer et al., 2017*). We expect the identification of *NDP* as a patterning

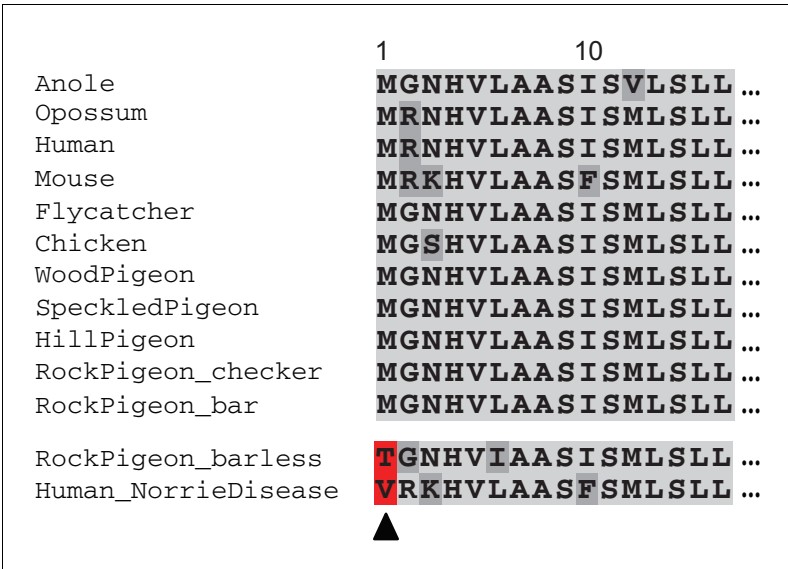

**Figure 4.** Barless pigeons have a nonsense mutation at the highly-conserved translation start site of *NDP*. Barless rock pigeons are homozygous for a nonsense mutation that truncates the amino terminus of *NDP* by 13 amino acids; the same start-codon position is affected by a mutation in two human families with hereditary blindness (red, bottom of alignments).

DOI: https://doi.org/10.7554/eLife.34803.018

gene to open new avenues of similar functional experiments to understand how pigment distribution is mediated.

## Signatures of introgression of the checker haplotype

Pigeon fanciers have long hypothesized that the checker pattern in the rock pigeon (*Columba livia*) resulted from a cross-species hybridization event with the speckled pigeon (*C. guinea, Figure 5D*), a species with a checker-like wing pattern (G. Hochlan, G. Young, personal communication) (*Hollander, 1983b*). We estimate that *C. livia* and *C. guinea* diverged 4–5 million years ago (MYA): columbid species (pigeons and doves) diverge from each other in mitochondrial cytochrome *b* nucleotide sequence at 1.96% per MY (*Weir and Schluter, 2008*), and *C. livia* and *C. guinea* differ at this gene by 8.0%. Divergence date estimates for these two species based on nuclear genome sequences range between 3.2 and 6.7 MYA (K.P.J., unpublished results). Despite this divergence time of several MY, inter-species crosses between *C. livia* and *C. guinea* can produce fertile hybrids (*Whitman, 1919*; *Irwin et al., 1936*; *Taibel, 1949*; *Miller, 1953*). Moreover, hybrid $F_1$ and backcross progeny between *C. guinea* and bar *C. livia* have checkered wings, much like *C. livia* with the *C* allele (*Whitman, 1919*; *Taibel, 1949*). *Taibel (1949)* showed that, although hybrid $F_1$ females were infertile, two more generations of backcrossing hybrid males to *C. livia* could produce checker offspring of both sexes that were fully fertile. In short, Taibel introgressed the checker trait from *C. guinea* into *C. livia* in just three generations.

To evaluate the possibility of an ancient introgression event, we sequenced an individual *C. guinea* genome to 33X coverage and mapped the reads to the *C. livia* reference assembly. We calculated four-taxon *D*-statistics ('ABBA-BABA' test; *Durand et al., 2011*) to test for deviations from expected sequence similarity between *C. guinea* and *C. livia*, using a wood pigeon (*C. palumbus*) genome as an outgroup (*Supplementary file 3*). In this case, the null expectation is that the *C* candidate region will be more similar between conspecific bar and checker *C. livia* than either will be to the same region in *C. guinea*. That is, the phylogeny of the candidate region should be congruent with the species phylogeny. However, we found that the *D*-statistic approaches one in the candidate region (n = 10 each for bar and checker *C. livia*), indicating that checker *C. livia* are more similar to *C. guinea* than to conspecific bar birds in this region (*Figure 5A*). The mean genome-wide *D*-statistic was close to zero (0.021), indicating that bar and checker sequences are more similar to each other throughout the genome than either one is to *C. guinea*.

This similarity between *C. guinea* and checker *C. livia* in the pattern candidate region was further supported by sequence analysis using HybridCheck (*Ward and van Oosterhout, 2016*). Outside of the candidate region, checker birds have a high sequence similarity to conspecific bar birds and low similarity to *C. guinea* (*Figure 5B*). Within the candidate region, however, this relationship shows a striking reversal, and checker and *C. guinea* sequences are most similar to each other. In addition, although the genome-wide *D*-statistic was relatively low, the 95% confidence interval (CI) was greater than zero (0.021 to 0.022), providing further evidence for one or more introgression events from *C. guinea* into checker and T-check genomes. Unlike in many checker and T-check *C. livia*, we did not find additional copies of the CNV region in *C. guinea*. This could indicate that the CNV expanded in *C. livia*, or that the CNV is present in a subset of *C. guinea* but has not yet been sampled. Taken together, these patterns of sequence similarity and divergence support the hypothesis that the candidate checker haplotype in rock pigeons originated by introgression from *C. guinea*.

While post-divergence introgression is an attractive hypothesis to explain the sequence similarity between checker *C. livia* and *C. guinea*, another formal possibility is that sequence similarity between these groups is due to incomplete lineage sorting. In an analogous example, light- and dark-pigmentation alleles of *tan* probably segregated in the ancestor of *Drosophila americana* and *D. novamexicana*, and the light allele subsequently became fixed in the latter species (*Wittkopp et al., 2009*). However, light and dark alleles continue to segregate in *D. americana*, and the light allele in this species has the same ancestral origin as the one that is fixed in *D. novamexicana*. Similarly, we wanted to test if the minimal checker haplotype might have been present in the last common ancestor of *C. guinea* and *C. livia*, but now segregates only in *C. livia*.

We measured nucleotide differences among different alleles of the minimal haplotype and compared these counts to polymorphism rates expected to accumulate over the 4-5 MY divergence time between *C. livia* and *C. guinea* (*Figure 5C*, purple bar, see Materials and methods).We found that polymorphisms between bar *C. livia* and *C. guinea* approached the number expected to accumulate

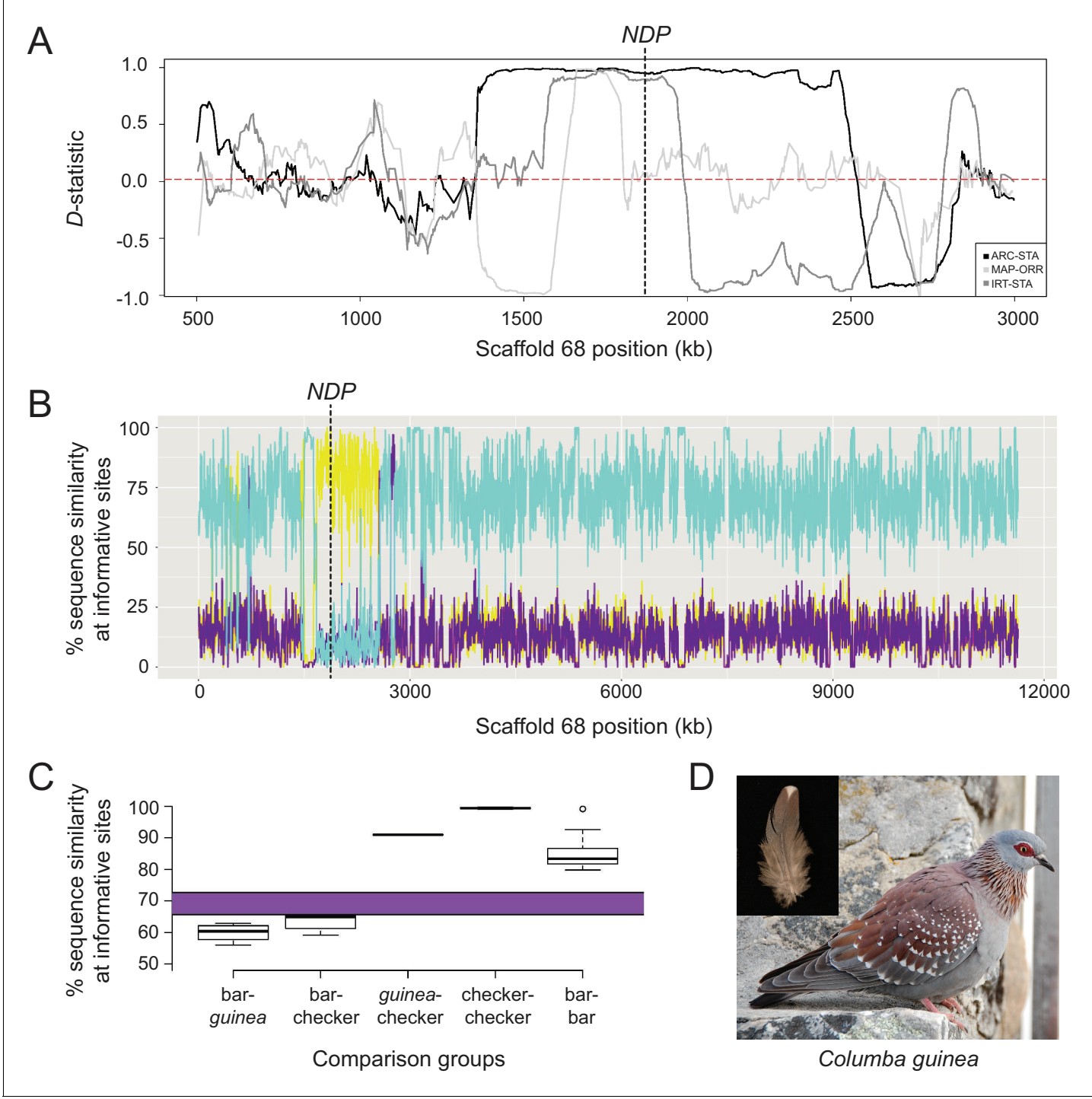

**Figure 5.** Signatures of introgression of the checker haplotype from *C. guinea* to *C. livia*. (A) ABBA-BABA test with *C. livia* (bar), *C. livia* (checker), *C. guinea*, and *C. palumbus* shows elevated *D*-statistic in the Scaffold 68 candidate region. Three representative ABBA-BABA tests are shown and dashed red line marks the genome-wide mean *D*-statistic for 10 × 10 different combinations of bar and checker birds (ARC-STA, MAP-ORR, IRT-STA are shown, where ARC, MAP, and IRT are checker samples and STA and ORR are bar samples; see Methods). (B) HybridCheck shows pairwise sequence similarity across informative sites of a sequence triplet. A representative triplet of bar (Fer_VA), checker (ARC), and *C. guinea* comparison is shown. Blue trace shows sequence similarity between bar and checker, purple trace shows similarity between bar and *C. guinea*, and yellow trace shows sequence similarity between checker and *C. guinea*. (C) Expected (purple bar) and observed proportion of shared segregating sites out of 4261 total SNPs in the minimal haplotype region for different pairwise comparisons between and among 16 bar, 11 checker, and 1 *C. guinea*. (D) Speckled pigeon (*Columba guinea*). Photo courtesy of Kjeuring (CC BY 3.0 license, https://creativecommons.org/licenses/by/3.0/legalcode). Photo cropped from 'speckled pigeon

*Figure 5 continued on next page*

*Figure 5 continued*

*Columba guinea* Table Mountain Cape Town,' https://en.wikipedia.org/wiki/Speckled_pigeon#/media/File:Speckledpigeon.JPG. Inset feather image by the authors.

DOI: https://doi.org/10.7554/eLife.34803.019

The following source data and figure supplement are available for figure 5:

**Source data 1.** Numbers of SNPs between different pairwise combinations of homozygous bar, checker, and *C. guinea* represented in *Figure 5C*.

DOI: https://doi.org/10.7554/eLife.34803.021

**Figure supplement 1.** Expected (purple bar) and observed proportion of shared segregating sites out of 1,458 SNPs in the minimal haplotype region for different pairwise comparisons between de novo genome assemblies from short-read resequencing data for bar, checker, and *C. guinea*.

DOI: https://doi.org/10.7554/eLife.34803.020

in this region in 4–5 MY (59.90% sequence similarity at segregating sites, SD = 2.6%, 1708 ± 109 mean SNPs, *Figure 5C*), but so did intraspecific comparisons between bar and checker *C. livia* (63.28%, SD = 2.3%, 1564 ± 99). In contrast, *C. guinea* and *C. livia* checker sequences had significantly fewer differences than would be expected to accumulate between the two species (90.96%, SD = 0.13%, 384 ± 6, p<2.2e-16, t-test). These results support an introgression event from *C. guinea* to *C. livia*, rather than a shared allele inherited from a common ancestor prior to divergence. Among 11 checker haplotype sequences, we found remarkably high sequence similarity (99.39%, SD = 0.18%, 26 ± 8 mean differences), corresponding to a haplotype divergence time of 89 ± 27 thousand years (KY), based on mutation rate.

The rock pigeon reference genome contains the checker haplotype, which could bias the discovery of SNPs in our resequenced genomes. We therefore performed de novo assemblies using Illumina shotgun reads from *C. guinea* and high-coverage bar and checker individuals, then compared nucleotide sequences in regions of the minimal haplotype where all three assemblies overlapped (92,199 of 102,909 bp, or 89.6%). We found similar patterns of divergence between the de novo assemblies and the resequenced genomes that were mapped to the reference, indicating that that SNP discovery was not heavily biased by our short-read mapping approach (*Figure 5—figure supplement 1*). Based on pairwise polymorphisms between the checker reference and the de novo checker assembly (11 differences), the haplotype divergence time is 42 KY. This figure is more recent than our estimate based on more individuals, but the key results are that both estimates are roughly 2 orders of magnitude more recent than the divergence time between species, and the similarity between checker and *C. guinea* sequences is characteristic of within-species rather than between-species variation.

Lastly, to date the putative introgression event(s), we estimated the age of the minimal checker haplotype based on the pattern of linkage disequilibrium decay (*Voight et al., 2006*). Using a recombination rate calculated for rock pigeon (*Holt et al., 2018*), the checker haplotype originated in *C. livia* between 429 and 857 years ago, assuming one to two generations per year. The corresponding 95% confidence intervals are 267 to 716 years ago assuming one generation per year and 534 to 1,432 years ago assuming two generations per year.

Together, these multiple lines of evidence support the hypothesis that the checker haplotype was introduced from *C. guinea* into *C. livia* after the domestication of the rock pigeon (~5000 years ago). The four-taxon *D*-statistic values approach one at the *NDP* locus (*Figure 5A*), indicating that checker *C. livia* is far more closely related to *C. guinea* than to bar *C. livia* at this locus. Additionally, the pairwise differences between *C. guinea* and checker haplotypes are incompatible with incomplete lineage sorting (*Figure 5C*), assuming a 4–5 MY species divergence time and no subsequent gene flow. The lack of single nucleotide diversity among checker haplotypes, with only 26 ± 8 mean differences and an estimated gene tree divergence of 89 KY, is unusually low for the diversity typically observed in large, free-living pigeon populations (*Shapiro et al., 2013*). The differences between the mutation-based (89 KY) and LD-based (0.4 to 0.9 KY) estimates of the checker haplotype age are an expected consequence of crossbreeding and artificial selection given that the former is an estimate of the age of the most recent common ancestor in the source population while the latter is a lower bound estimate for the date of introgression. Inconsistencies of this magnitude are unexpected in the absence of introgression. Additionally, the genome-wide *D*-statistic comparing *C. guinea* and bar to *C. guinea* and checker is low but significantly greater than 0, indicating that gene flow from *C. guinea* to checker has been higher than from *C. guinea* to bar throughout the genome.

Notably, the non-zero *D*-statistic result holds when the *NDP* locus is excluded from this calculation. These results are expected if the checker haplotypes were recently introduced into *C. livia* by pigeon breeders, and interbreeding between checker and bar populations has not been completely random. Consistent with this expectation, non-random mating is observed in feral populations, and pigeon breeders often impose color pattern selection on their birds (*Darwin, 1868*; *Burley, 1977*; *1981*; *Johnston and Johnson, 1989*; *National_Pigeon_Association, 2010*).

Finally, the upper bound of the LD-based age estimate of the checker haplotype of 1,432 years ago indicates that checker haplotype was introduced into *C. livia* well after the domestication of rock pigeons. Because the ranges of *C. livia* and *C. guinea* overlap in northern Africa (*del Hoyo et al., 2017*), it is possible that introgression events occurred in free-living populations. However, the more likely explanation is that *C. guinea* haplotypes were introduced into *C. livia* by pigeon breeders. Once male hybrids are generated, this can be accomplished in just a few generations (*Taibel, 1949*).Thus, humans might have intentionally selected this phenotype, which is linked to life history traits that are advantageous in urban environments, and then built ideal urban habitats for them to thrive (*Jerolmack, 2008*).

## Introgression and pleiotropy

Adaptive traits can arise through new mutations or standing variation within a species, and a growing number of studies point to adaptive introgressions among vertebrates and other organisms (*Hedrick, 2013*; *Martin and Orgogozo, 2013*; *Harrison and Larson, 2014*; *Zhang et al., 2016*). In some cases, introgressed loci are associated with adaptive traits in the receiving species, including high-altitude tolerance in Tibetan human populations from Denisovans (*Huerta-Sánchez et al., 2014*), resistance to anticoagulant pesticides in the house mouse from the Algerian mouse (*Song et al., 2011*; *Liu et al., 2015*), and beak morphology among different species of Darwin's finches (*Lamichhaney et al., 2015*). Among domesticated birds, introgressions are responsible for skin and plumage color traits in chickens and canaries, respectively (*Eriksson et al., 2008*; *Lopes et al., 2016*). Alleles under artificial selection in a domesticated species can be advantageous in the wild as well, as in the introgression of dark coat color from domestic dogs to wolves (*Anderson et al., 2009*) (however, color might actually be a visual marker for an advantageous physiological trait conferred by the same allele; *Coulson et al., 2011*).

In this study, we identified a putative introgression into *C. livia* from *C. guinea* that is advantageous both in artificial (selection by breeders) and free-living urban environments (sexual and natural selection). A change in plumage color pattern is an immediately obvious phenotypic consequence of the checker allele, yet other traits are linked to this pigmentation pattern. For example, checker and T-check pigeons have longer breeding seasons, up to year-round in some locations (*Lofts et al., 1966*; *Murton et al., 1973*), and *C. guinea* breeds year-round in most of its native range as well (*del Hoyo et al., 2017*). Perhaps not coincidentally, *NDP* is expressed in the gonad tissues of adult *C. livia* (*MacManes et al., 2017*) and the reproductive tract of other amniotes (*Paxton et al., 2010*). Abrogation of expression or function of *NDP* or its receptor *FZD4* is associated with infertility and gonad defects (*Luhmann et al., 2005*; *Kaloglu et al., 2011*; *Ohlmann et al., 2012*; *Ohlmann and Tamm, 2012*). Furthermore, checker and T-check birds deposit less fat during normally reproductively quiescent winter months. In humans, expression levels of *FZD4* and the co-receptor *LRP5* in adipose tissue respond to varying levels of insulin (*Karczewska-Kupczewska et al., 2016*), and *LRP5* regulates the amount and location of adipose tissue deposition (*Loh et al., 2015*; *Karczewska-Kupczewska et al., 2016*). Therefore, based on its reproductive and metabolic roles in pigeons and other amniotes, *NDP* is a viable candidate not only for color pattern variation, but also for the suite of other traits observed in free-living (feral and wild) checker and T-check pigeons. Indeed, the potential pleiotropic effects of *NDP* raise the possibility that reproductive output and other physiological advantages are secondary or even primary targets of selection, and melanistic phenotypes are honest genetic signals of a cluster of adaptive traits controlled by a single locus.

Adaptive *cis*-regulatory change is also an important theme in the evolution of vertebrates and other animals (*Shapiro et al., 2004*; *Miller et al., 2007*; *Wray, 2007*; *Carroll, 2008*; *Chan et al., 2010*; *Wittkopp and Kalay, 2011*; *O'Brown et al., 2015*; *Signor and Nuzhdin, 2018*). This theme is especially prominent in studies of color variation in *Drosophila*, in which regulatory variation impacts both the type and pattern of pigments on the body and wings (*Gompel et al., 2005*; *Prud'homme et al., 2006*; *Rebeiz et al., 2009*). In some cases, the evolution of multiple regulatory

elements of the same gene can fine-tune phenotypes, such as mouse coat color and trichome distribution in fruit flies (*McGregor et al., 2007*; *Linnen et al., 2013*). In cases of genes that have multiple developmental roles, introgression can result in the simultaneous transfer of multiple advantageous traits (*Rieseberg, 2011*). The potential role of *NDP* in both plumage and physiological variation in pigeons could represent a striking example of pleiotropic regulatory effects.

Wing pigmentation patterns that resemble checker are present in many wild species within and outside of Columbidae including *Patagioenas maculosa* (Spot-winged pigeon), *Spilopelia chinensis* (Spotted dove), *Geopelia cuneata* (Diamond dove), *Gyps rueppelli* (Rüppell's vulture), and *Pygiptila stellaris* (Spot-winged antshrike). Based on our results in pigeons, *NDP* and its downstream targets can serve as initial candidate genes to ask whether similar molecular mechanisms generate convergent patterns in other species.

## Materials and methods

### Ethics statement
Animal husbandry and experimental procedures were performed in accordance with protocols approved by the University of Utah Institutional Animal Care and Use Committee (protocols 10–05007, 13–04012, and 16–03010).

### DNA sample collection and extraction
Blood samples were collected in Utah at local pigeon shows, at the homes of local pigeon breeders, from pigeons in the Shapiro lab, and from ferals that had been captured in Salt Lake City, Utah. Photos of each bird were taken upon sample collection for our records and for phenotype verification. Tissue samples of *C. rupestris*, *C. guinea*, and *C. palumbus* were provided by the University of Washington Burke Museum, Louisiana State University Museum of Natural Science, and Tracy Aviary, respectively. Breeders outside of Utah were contacted by email or phone to obtain feather samples. Breeders were sent feather collection packets and instructions, and feather samples were sent back to the University of Utah along with detailed phenotypic information. Breeders were instructed to submit only samples that were unrelated by grandparent. DNA was then extracted from blood, tissue, and feathers as previously described (*Stringham et al., 2012*).

### Determination of color and pattern phenotype of adult birds
Feather and color phenotypes of birds were designated by their respective breeders. Birds that were raised in our facility at the University of Utah or collected from feral populations were assigned a phenotype using standard references (*Levi, 1986*; *Sell, 2012*).

### Genomic analyses
BAM files from a panel of previously resequenced birds were combined with BAM files from eight additional barless birds, 23 bar and 23 checker birds (22 feral, 24 domestics), a single *C. guinea*, and a single *C. palumbus*. SNVs and small indels were called using the Genome Analysis Toolkit (Unified Genotyper and LeftAlignAnd TrimVariants functions, default settings; *McKenna et al., 2010*). Variants were filtered as described previously (*Domyan et al., 2016*) and the subsequent variant call format (VCF) file was used for pFst and ABBA-BABA analyses as part of the VCFLIB software library (https://github.com/vcflib) and VAAST (*Yandell et al., 2011*) as described previously (*Shapiro et al., 2013*).

pFst was first performed on whole-genomes of 32 bar and 27 checker birds. Some of the checker and bar birds were sequenced to low coverage (~1X), so we were unable to confidently define the boundaries of the shared haplotype. To remedy this issue, we used the core of the haplotype to identify additional bar and checker birds from a set of birds that had already been sequenced to higher coverage (*Shapiro et al., 2013*). These additional birds were not included in the initial scan because their wing pattern phenotypes were concealed by other color and pattern traits that are epistatic to bar and check phenotypes. For example, the recessive red (*e*) and spread (*S*) loci produce a uniform pigment over the entire body, thereby obscuring any bars or checkers (*Van Hoosen Jones, 1922*; *Hollander, 1938a*; *Sell, 2012*; *Domyan et al., 2014*). Although the major wing pattern is not visible in these birds, the presence or absence of the core checker haplotype allowed us to

characterize them as either bar or checker/T-check. We then re-ran pFst using 17 bar and 24 checker/T-check birds with at least 8X mean read depth coverage (*Figure 1B*) and found a minimal shared checker haplotype of ~100 kb (Scaffold 68 position 1,702,691–1,805,600), as defined by haplotype breakpoints in a homozygous checker and a homozygous bar bird (NCBI BioSamples SAMN01057561 and SAMN01057543, respectively; BioProject PRJNA167554). pFst was also used to compare the genomes of 32 bar and nine barless birds. New sequence data for *C. livia* are deposited in the NCBI SRA database under BioProject PRJNA428271 with the BioSample accession numbers SAMN08286792- SAMN08286844. New sequence data for *C. guinea* and *C. palumbus* are deposited in the NCBI SRA database under accession numbers SRS1416880 and SRS1416881, respectively.

## Pedigree of an $F_2$ intercross segregating checker and bar

We genotyped and phenotyped a laboratory intercross that segregates bar and checker patterns in the $F_2$ generation. We generated a pedigree from this family for $F_2$ individuals whose phenotypes we could identify as bar or checker (n = 62). We could not determine bar or checker phenotypes for all individuals because other pigment patterns that epistatically mask bar and checker – almond (*St* locus), spread (*S*), and recessive red (*E*) – are also segregating in the cross. $F_2$ individuals were excluded from the analysis if they had one of these masking phenotypes, but $F_1$ parents were retained if they produced $F_2$ offspring with checker or bar phenotypes. We used primers that amplify within the minimal haplotype (AV17 primers, see *Supplementary file 1*) to genotype all $F_2$ individuals, their $F_1$ parents (n = 26), and the founders (n = 4) by Sanger sequencing for the checker haplotype to assess whether the checker haplotype segregated with wing pattern phenotype.

## CNV breakpoint identification and read depth analysis

The approximate breakpoints of the CNV region were identified at Scaffold 68 positions 1,790,000 and 1,805,600 using WHAM in resequenced genomes of homozygous bar or checker birds with greater than 8x coverage (*Kronenberg et al., 2015*). For 12 bar, seven checker, and 2 T-check resequenced genomes, Scaffold 68 gdepth files were generated using VCFtools (*Danecek et al., 2011*). Two subset regions were specified: the first contained the CNV and the second was outside of the CNV and was used for normalization (positions 1,500,000–2,000,000 and 800,000–1,400,000, respectively). Read depth in the CNV was normalized by dividing read depth in this region by the average read depth from the second (non-CNV) region, then multiplying by two to normalize for diploidy.

## Taqman assay for copy number variation

Copy number variation was estimated using a custom Taqman Copy Number Assay (assay ID: cnvtaq1_CC1RVED; Applied Biosystems, Foster City, CA) for 93 birds phenotyped by wing pigment pattern category and 89 birds whose pigmentation was quantified by image analysis. After DNA extraction, samples were diluted to 5 ng/μL. Samples were run in quadruplicate according to the manufacturer's protocol.

## Quantification of pigment pattern phenotype

At the time of blood sample collection, the right wing shield was photographed (RAW format images from a Nikon D70 or Sony a6000 digital camera). Using Photoshop software (Adobe Systems, San Jose, CA), the wing shield including the bar (on the secondary covert feathers) was isolated from the original RAW file. Images were adjusted to remove shadows and the contrast was set to 100%. The isolated adjusted wing shield image was then imported into ImageJ (imagej.nih.gov/) in JPEG format. Image depth was set to 8-bit and we then applied the threshold command. Threshold was further adjusted by hand to capture checkering and particles were analyzed using a minimum pixel size of 50. This procedure calculated the area of dark plumage pigmentation on the wing shield. Total shield area was calculated using the Huang threshold setting and analyzing the particles as before (minimum pixel size of 50). The dark area particles were divided by total wing area particles, and then multiplied by 100 to get the percent dark area on the wing shield. Measurements were done in triplicate for each bird, and the mean percentages of dark area for each bird were used to test for associations between copy number and phenotype using a non-linear least squares regression.

## qRT-PCR analysis of gene expression

Two secondary covert wing feathers each from the wing shields of 8 bar, seven checker, and 8 T-check birds were plucked to stimulate feather regeneration for qRT-PCR experiments. Nine days after plucking, regenerating feather buds were removed, the proximal 5 mm was cut longitudinally, and specimens were stored in RNAlater (Qiagen, Valencia, CA) at 4°C for up to three days. Next, collar cells were removed, RNA was isolated, and mRNA was reverse-transcribed to cDNA as described previously (*Domyan et al., 2014*). Intron-spanning primers (see *Supplementary file 1*) were used to amplify each target using a CFX96 qPCR instrument and iTaq Universal Syber Green Supermix (Bio-Rad, Hercules, CA). Samples were run in duplicate and normalized to β-actin. The mean value was determined and results are presented as mean ± S.E. for each phenotype. Results were compared using a Wilcoxon Rank Sum test and expression differences were considered statistically-significant if p<0.05.

## Allele-specific expression assay

SNPs in *NDP* and *EFHC2* were identified as being diagnostic of the bar or checker/T-check haplotypes from resequenced birds. Heterozygous birds were identified by Sanger sequencing in the minimal checker haplotype region (AV17 primers, see *Supplementary file 1*). Twelve checker and T-check heterozygous birds were then verified by additional Sanger reactions (AV54 for *NDP* and AV97 for *EFHC2*, see *Supplementary file 1*) to be heterozygous for the diagnostic SNPs in *NDP* and *EFHC2*. PyroMark Custom assays (Qiagen, Valencia, CA) were designed for each SNP using the manufacturer's software (*Supplementary file 1*). Pyrosequencing was performed on gDNA derived from blood and cDNA derived from collar cells from 9 day regenerating wing shield feathers using a Pyro-Mark Q24 instrument (Qiagen, Valencia, CA). Additional pyrosequencing was performed for 9 of the 12 of the original birds from 9 day regenerating dorsal and tail feathers following the same protocol. Signal intensity ratios from the cDNA samples were normalized to the ratios from the corresponding gDNA samples to control for bias in allele amplification. Normalized ratios were analyzed by Wilcoxon Rank Sum tests. We compared the expression ratios of 1-copy checker:bar to 4-copy checker:bar to determine whether additional copies of the CNV were associated with higher checker:bar allele expression. We also compared 1-copy checker:bar expression ratios and four copy checker:bar expression ratios to a 1:1 ratio (equal expression of both alleles) using the Wilcoxon Rank Sum test to determine whether the measured checker:bar ratios were significantly different from the null hypothesis of equal expression of bar and checker alleles. The 2-copy checker:bar ratio was not compared in these analyses because there was only one sample. Allele expression ratios were analyzed together for 1, 2, and 4-copies using a glm regression to determine whether CNV copy number was associated with increased checker allele expression. Results were considered significant if p<0.05.

## Enhancer sequence search

VISTA (https://enhancer.lbl.gov/) (*Visel et al., 2007*) and REPTILE (*He et al., 2017*) enhancer data-sets were mapped to the pigeon reference genome using bwa-mem (*Li and Durbin, 2009*). BAM output files were filtered for high quality orthologous regions and further filtered for alignments within the minimal checker haplotype on Scaffold 68 (*Supplementary file 2*).

## NDP genotyping and alignments

*NDP* exons were sequenced using primers in *Supplementary file 1*. Primers pairs were designed using the rock pigeon reference genome (Cliv_1.0) (*Shapiro et al., 2013*). PCR products were purified using a QIAquick PCR purification kit (Qiagen, Valencia, CA) and Sanger sequenced. Sequences from each exon were then edited for quality with Sequencher v.5.1 (GeneCodes, Ann Arbor, MI). Sequences were translated and aligned with SIXFRAME and CLUSTALW in SDSC Biology Work-bench (http://workbench.sdsc.edu). Amino acid sequences outside of Columbidae were downloaded from Ensembl (www.ensembl.org).

## *D*-statistic calculations

Whole genome ABBA-BABA (https://github.com/vcflib) was performed on 10 × 10 combinations of bar and checker (*Supplementary file 3*) birds in the arrangement: bar, checker, *C. guinea*, *C. palumbus*. VCFLIB (https://github.com/vcflib) was used to smooth raw ABBA-BABA results in 1000 kb or

100 kb windows for whole-genome or Scaffold 68 analyses respectively. For each 10 × 10 combination. We calculated the average *D* statistic across the genome. These were then averaged to generate a whole genome average of *D* = 0.0212, marked as the dotted line in *Figure 5A*. Confidence intervals were generated via moving blocks bootstrap (*Kunsch, 1989*). Block sizes are equal to the windows above, with *D*-statistic values resampled with replacement a number of times equal to the number of windows in a sample. In *Figure 5A*, three representative ABBA-BABA tests are shown for different combinations of bar and checker birds. The checker and bar birds used in each representative comparison are: ARC-STA, SRS346901 and SRS346887; MAP-ORR, SRS346893 and SRS346881; IRT-STA, SRS346892 and SRS346887 respectively. ARC, MAP, and IRT are homozygous for the checker haplotype. STA and ORR are homozygous for the bar haplotype.

## Haplotype phasing and HybridCheck analysis

VCF files containing Scaffold 68 genotypes for 16 bar, 11 homozygous checker, and 1 *C. guinea* were phased using Beagle version 3.3 (*Browning and Browning, 2007*). VCFs were then converted to fasta format using vcf2fasta in vcf-lib (https://github.com/vcflib). HybridCheck (*Ward and van Oosterhout, 2016*) (https://github.com/Ward9250/HybridCheck) was run to visualize pairwise sequence similarities between trios of bar, checker, and *C. guinea* sequences across Scaffold 68 using default settings.

## Pairwise SNP comparisons

Phased VCF files for 16 homozygous bar, 11 homozygous checker, and 1 *C. guinea* were subsetted to the minimal checker haplotype region (positions 1,702,691–1,805,600) with tabix (*Li, 2011*). The vcf-compare software module (VCFtools, (*Danecek et al., 2011*) was used to run pairwise comparisons between bar, checker, and *C. guinea* birds (176 bar-checker, 16 bar-guinea, and 11 checker-guinea comparisons) as well as among bar and checker birds (120 bar-bar and 55 checker-checker comparisons). The total number of differences for each group was compared to the number of differences that are expected to accumulate during a 4–5 MY divergence time in a 102,909 bp region (the size of the minimal checker haplotype) with the mutation rate μ = 1.42e-9 (*Shapiro et al., 2013*) using the coalescent equation: Time= #SNPs/(2xμx length of the region). The observed pairwise differences and the expected number of differences were evaluated with two-sample t-tests and all groups were considered statistically different from the 4–5 MY expectation (1169.05–1461.31). There were 4261 total segregating sites in the minimal haplotype region between all birds used for pairwise comparisons. Means and standard deviations for each group were calculated in R (*R_Development_Core_Team, 2008*).

## SNP comparisons in de novo assemblies of bar, checker, and *C. guinea* genomes

To ensure that SNP calling was not biased by using a reference that has the checker haplotype, we performed de novo assemblies of one bar (SRS346895), one checker (SRS346878), and one *C. guinea* (SRS1416880) individual using CLC Genomics Workbench (Qiagen, Valencia, CA). These *C. livia* individuals were chosen because they had the highest genome-wide mean read depth coverage for each phenotype at 14X (bar) and 15X (checker; the *C. guinea* sample was sequenced to 33X). Whole-genome assemblies were mapped to the reference genome and variants (single nucleotide variants, structural variants, indels) were called by SMARTIE-SV (https://github.com/zeeev/smartie-sv), which uses the BLASR aligner (*Chaisson and Tesler, 2012*), using default parameters. We identified regions where all three new assemblies intersected with the reference assembly. We then counted SNPs across the minimal haplotype where all three assemblies intersected (92,199 of 102,909 bp; 12 intersecting contigs ranging in length from 678 to 21565 bp, median = 5047.5).

Variants identified in the de novo assemblies for checker, bar, or *C. guinea* individuals were manually filtered to remove variants where the alternate allele was 'N' or a series of 'N' base pairs. Variants spanning multiple base pairs in each individual file were identified and manually split into multiple single nucleotide polymorphisms. Filtered and split tab-delimited variant calls between each de novo assembly and the reference genome were read into R v.3.3.2 (*R_Development_Core_Team, 2008*). For each variant call file, the start position was extracted. Pairwise comparisons of positions for checker, bar, and *C. guinea* de novo assemblies were made using the 'setdiff'

command to generate lists of variants that were only observed in one individual out of any given pair (checker vs. bar, checker vs. *C. guinea*, bar vs. *C. guinea*). These lists of positions were then used to subset the original variant call files and assemble lists of pairwise differences. For example, SNPs that differ between checker and bar would include variants that differ from the reference in checker, but not bar, plus variants that differ from the reference in bar, but not checker.

Additionally, the 'intersect' command was used to identify variants in multiple de novo assemblies. For variants that appeared in more than one de novo assembly, alternative alleles for each assembly were compared. In the majority of cases, both de novo assemblies showed the same alternative allele, and thus did not differ from one another.

We found 1458 total SNP positions based on comparison of the three de novo assemblies. In the comparison described above and shown in *Figures 5C*, 362 SNPs were identified in the same region. This higher number of SNPs was driven by the much larger sample size and haplotype diversity among the 16 bar birds.

## Transcript amplification of barless allele of *NDP*

In order to determine whether the barless allele of *NDP* is transcribed and persists in collar cells, or is degraded (e.g., by non-sense mediated decay), we designed a PCR assay to amplify *NDP* mRNA transcripts. Feathers from four barless, 2 bar, two checker, and 2 T-check birds were plucked to stimulate regeneration. We then harvested regenerated feathers after 9 days, extracted RNA from collar cells, and synthesized cDNA as described above. We then generated amplicons from each sample using intron-spanning primers (AV200 primers, see *Supplementary file 1*). Primers were anchored in the exon containing the barless start-codon mutation and the exon 3' to it, so this assay tested for both the presence of transcripts and consistent splicing among alleles and phenotypes.

## *EFHC2* alignments

*EFHC2* exonic sequences from resequenced homozygous bar (n = 16), homozygous check or T-check (n = 11), and barless (n = 9) *Columba livia; C. rupestris* (n = 1); *C. guinea* (n = 1); and *C. palumbus* (n = 1) were extracted using the IGV browser (*Thorvaldsdóttir et al., 2013*). Exon sequences for each group were translated using SIXFRAME in SDSC Biology Workbench (http://workbench.sdsc.edu). Peptide sequences were then aligned to EFHC2 amino acid sequences from other species downloaded from ensembl (http://www.ensembl.org) using CLUSTALW (*Thompson et al., 1994*) in SDSC Biology Workbench. Exon sequences from additional *C. livia* (n = 17 checker or T-check, and n = 14 bar) and *C. guinea* (n = 5) birds were determined by Sanger sequencing.

## Recombination rate estimation

Recombination frequency estimates were generated from a genetic map based an F2 cross of two divergent *C. livia* breeds, a Pomeranian Pouter and a Scandaroon (*Domyan et al., 2016*). Briefly, for genetic map construction, genotyping by sequencing (GBS) data were generated, trimmed, and filtered as described (*Domyan et al., 2016*), then mapped to the pigeon genome assembly (*Holt et al., 2018*) using Bowtie2 (*Langmead and Salzberg, 2012*). Genotypes were called using Stacks (*Catchen et al., 2011*), and genetic map construction was performed using R/qtl (www.rqtl.org) (*Broman et al., 2003*). Pairwise recombination frequencies were calculated for all markers based on GBS genotypes. Within individual scaffolds, markers were filtered to remove loci showing segregation distortion (Chi-square, p<0.01) or probable genotyping error. Specifically, markers were removed if dropping the marker led to an increased LOD score, or if removing a non-terminal marker led to a decrease in length of >10 cM that was not supported by physical distance. Individual genotypes with error LOD scores > 5 (*Lincoln and Lander, 1992*) were also removed. Pairwise recombination frequencies for markers flanking the candidate region that were retained in the final linkage map were used to estimate the age of the introgression event between *C. guinea* and *C. livia* (Scaffold 68, marker positions 1,017,014 and 1,971,666; *Supplementary file 4*).

## Minimal haplotype age estimation

The minimal haplotype age was estimated following *Voight et al. (2006)*. We assume a star-shaped phylogeny, in which all samples with the minimal haplotype are identical to the nearest recombination event, and differ immediately beyond it. Choosing a variant in the center of the minimal

haplotype, we calculated EHH, and estimated the age using the largest haplotype with a probability of homozygosity just below 0.25. Note that

$$\Pr[homoz] = e^{-2rg}$$

where r is the genetic map distance, and g is the number of generations since introgression / onset of selection. Therefore

$$g = -\frac{100 \log(\Pr[homoz])}{2r}$$

The confidence interval around g was estimated assuming

$$N \sim Binom(n = 22, \ p = 0.204)$$

Here, N is a binomially distributed random variable for the number of samples that have not recombined to a map distance equal to 2 r. Then, Pr[homoz]=N/22. The probability that a sample has no recombination event within 2 r of the focal SNP is p = (Pr[homoz | left]+Pr[homoz | right])/2 is derived from the data. Both left and right of the focal SNP we chose the end of the haplotype at the first SNP which brought Pr[homoz]<0.25.

## Acknowledgements

We thank past and present members of the Shapiro lab for assistance with sample collection and processing; members of the Utah Pigeon Club and National Pigeon Association for sample contributions; and Gene Hochlan, Gary Young, and Robert Mangile for critical discussions and advice. Mr. Hochlan also generously provided feather samples from *C. guinea* that helped us assess feasibility of the introgression study. We thank Fred Adler, Brett Boyd, Elena Boer, Alexa Davis, Cassandra Garner, Robert Greenhalgh, JJ Horns, Christopher Leonard, Jon Seger, Scott Villa, and Eric Domyan for technical assistance and advice. Dale Clayton and Sarah Bush generously provided field-collected tissue samples of *C. guinea* and *C. palumbus* for whole-genome sequencing. We thank Safari West (Santa Rosa, CA), the Louisiana State University Museum of Natural Science, and the University of Washington Burke Museum for additional *C. guinea* tissue samples (museum accessions 95045 JK 00 179, 101559 BCA 523, and 119004 EEM 979). This work was supported by the National Science Foundation (CAREER DEB-1149160 to MDS; GRF 1256065 to AIV and RB; and DEB-1342604 to KPJ) the National Institutes of Health (R01GM115996 to MDS, R01GM104390 to MY; fellowships T32GM007464 to ZK and R25CA057730 to RJB); and the Society for Developmental Biology (Choose Development fellowship to RM.). ETM is a fellow of the Jane Coffin Childs Memorial Fund for Medical Research. The funders had no role in study design, data collection and analysis, decision to publish, or preparation of the manuscript. We acknowledge a computer time allocation from the Center for High Performance Computing at the University of Utah and thank Phase Genomics for the use of computing resources.

## Additional information

### Funding

| Funder | Grant reference number | Author |
| --- | --- | --- |
| National Institute of General Medical Sciences | R01GM115996 | Michael D Shapiro |
| National Science Foundation | CAREER DEB-1149160 | Michael D Shapiro |
| National Institute of General Medical Sciences | R01GM104390 | Mark Yandell |
| National Science Foundation | GRF 1256065 | Anna I Vickrey Rebecca Bruders |
| National Science Foundation | DEB-1342604 | Kevin P Johnson |

| National Institute of General Medical Sciences | T32GM007464 | Zev Kronenberg |
| National Cancer Institute | R25CA057730 | Ryan J Bohlender |
| Jane Coffin Childs Memorial Fund for Medical Research | Postdoctoral Fellowship | Emily T Maclary |
| Society for Developmental Biology | Choose Development Fellowship | Raquel Maynez |

The funders had no role in study design, data collection and interpretation, or the decision to submit the work for publication.

## Author contributions

Anna I Vickrey, Conceptualization, Formal analysis, Funding acquisition, Investigation, Visualization, Methodology, Writing—original draft, Writing—review and editing; Rebecca Bruders, Formal analysis, Funding acquisition, Visualization, Writing—review and editing; Zev Kronenberg, Software, Formal analysis, Funding acquisition, Investigation, Methodology, Writing—review and editing; Emma Mackey, Formal analysis, Investigation, Review and approval of final manuscript; Ryan J Bohlender, Software, Formal analysis, Funding acquisition, Visualization, Writing—original draft, Writing—review and editing; Emily T Maclary, Formal analysis, Funding acquisition, Investigation, Writing—original draft, Writing—review and editing; Raquel Maynez, Formal analysis, Funding acquisition, Investigation, Review and approval of final manuscript; Edward J Osborne, Data curation, Software, Formal analysis, Validation; Kevin P Johnson, Resources, Data curation, Funding acquisition, Review and approval of final manuscript; Chad D Huff, Software, Formal analysis, Supervision, Writing—original draft, Writing—review and editing; Mark Yandell, Conceptualization, Resources, Software, Supervision, Funding acquisition, Methodology, Project administration, Review and approval of final manuscript; Michael D Shapiro, Conceptualization, Resources, Formal analysis, Supervision, Funding acquisition, Investigation, Methodology, Writing—original draft, Project administration, Writing—review and editing

## Author ORCIDs

Ryan J Bohlender (iD) http://orcid.org/0000-0002-6347-0494
Michael D Shapiro (iD) http://orcid.org/0000-0003-2900-4331

## Ethics

Animal experimentation: Animal experimentation: This study was performed in accordance with the recommendations in the Guide for the Care and Use of Laboratory Animals of the National Institutes of Health. All of the animals were handled and housed according to approved University of Utah institutional animal care and use committee (IACUC) protocols 10-05007, 13-04012, and 16-03010.

## Decision letter and Author response

Decision letter https://doi.org/10.7554/eLife.34803.036
Author response https://doi.org/10.7554/eLife.34803.037

# Additional files

## Supplementary files

• Supplementary file 1. Primer sequences used in this study.
DOI: https://doi.org/10.7554/eLife.34803.022

• Supplementary file 2. VISTA and REPTILE enhancer mapping hits on Scaffold 68.
DOI: https://doi.org/10.7554/eLife.34803.023

• Supplementary file 3. Combinations of bar and checker bird identifiers used to generate whole-genome *D*-statistics.
DOI: https://doi.org/10.7554/eLife.34803.024

• Supplementary file 4. Genotypes in the candidate region of individual birds in an $F_2$ cross of domestic pigeons used to estimate recombination frequency.
DOI: https://doi.org/10.7554/eLife.34803.025

• Transparent reporting form
DOI: https://doi.org/10.7554/eLife.34803.026

## Data availability

New sequence data for *C. livia* are deposited in the NCBI SRA database under BioProject PRJNA428271 with the BioSample accession numbers SAMN08286792- SAMN08286844. New sequence data for *C. guinea* and *C. palumbus* are deposited in the NCBI SRA database under accession numbers SRS1416880 and SRS1416881, respectively. Source data files are provided for Figures 2, 3, and 5.

The following datasets were generated:

| Author(s) | Year | Dataset title | Dataset URL | Database, license, and accessibility information |
|---|---|---|---|---|
| AI Vickrey, R Bruders, Z Kronenberg, E Mackey, RJ Bohlender, E Maclary, EJ Osborne, KP Johnson, CD Huff, M Yandell, MD Shapiro | 2018 | Genetics of wing pigment patterning in rock pigeon | https://www.ncbi.nlm.nih.gov/sra/SRP127865 | Publicly available at NCBI BioProject (accession no. PRJNA428271) |
| KP Johnson, B Boyd | 2018 | Whole-genome short-read sequencing of Columba guinea | https://www.ncbi.nlm.nih.gov/sra/SRS1416880 | Publicly available at NCBI Sequence Read Archive (accession no. SRS1416880) |
| KP Johnson, B Boyd | 2018 | Whole-genome short-read sequencing of Columba palumbus | https://www.ncbi.nlm.nih.gov/sra/SRS1416881 | Publicly available at NCBI Sequence Read Archive (accession no. SRS1416881) |

The following previously published dataset was used:

| Author(s) | Year | Dataset title | Dataset URL | Database, license, and accessibility information |
|---|---|---|---|---|
| Shapiro MD, Kronenberg Z, Li C, Domyan ET, Pan H, Campbell M, Tan H, Huff CD, Hu H, Vickrey AI, Nielsen SC, Stringham SA, Willerslev E, Gilbert MT, Yandell M, Zhang G, Wang J | 2013 | Whole genome sequencing of rock pigeon | http://www.ncbi.nlm.nih.gov/sra/?term=SRA054391 | Publicly available at NCBI Sequence Read Archive (accession no. SRA054391) |

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
