## [Decision Letter]

Thank you for submitting your article "Protein-coding variation and introgression of regulatory alleles drive plumage pattern diversity in the rock pigeon" for consideration by *eLife*. Your article has been favorably evaluated by Patricia Wittkopp (Senior Editor) and three reviewers, one of whom is a member of our Board of Reviewing Editors. The following individual involved in review of your submission has agreed to reveal his identity: Ricardo Mallarino (Reviewer #2).

The reviewers have discussed the reviews with one another and the Reviewing Editor has drafted this decision to help you prepare a revised submission.

Summary:

Vickrey et al. examine the molecular basis and evolutionary history of the *C* locus allelic series in the context of its involvement in plumage patterning in pigeons. Using a combination of genomic approaches and gene expression assays they find that wing pattern variation maps to both *cis*-regulatory elements (copy number variation) and the coding region of the *NDP* gene, a locus that has previously been implicated in pigmentation and neurological defects. Moreover, they show that the derived allele carrying the CNV originated in a different species, providing a nice example of adaptive introgression. The topic is fascinating, the methods are well executed, and the results are interesting.

Essential revisions:

1) There is no linkage/QTL mapping of the color pattern phenotypes. The authors note the prior work on Mendelian genetics, which suggests that there may be useful pedigrees in which alleles at the *C* locus segregate. The results of association mapping are striking, but there is a step that has been skipped here, which is showing that the *C* locus phenotypes actually co-segregate with this genomic region. Given the results presented in the study, genetic mapping is not strictly necessary, but this would be a nice addition which should be straightforward.

2) One of the remarkable aspects of color patterning is that there can be such tight spatial control of the mechanisms driving the phenotype. For example, the *C* allele seems to affect only the wing. Can the authors comment on any possible explanation for this? What happens with *NDP* expression elsewhere in the body? Is it even expressed? If not, why is it restricted to the wing? What is known about the source of this secreted factor? Incorporating some of these elements into the discussion, even if its speculative, would improve the manuscript.

3) The CNV description, subsection “A copy number variant is associated with melanistic wing patterns”, needs to be clarified. Our understanding of the results presented is that while an association exists, there are mismatches – checker and T-check birds that do not have additional copies. So, in terms of the CNV at least, they are like bar? Combined with the absence of the CNV in *C. guinea*, the results presented argue that the CNV is another associated marker but not causal. Therefore, the current, strong focus on the CNV may be not be warranted and one is left wondering about the possible mechanisms by which the CNV increases NDP expression (see also point #2). Did the authors examine the sequence to look for possible transcription factor binding sites? This may provide insights into why this is a spatially restricted pattern (e.g., a transcription factor that is locally secreted may bind to the CNV).

4) There are two potential issues with the divergence estimates.a) The estimates may be inaccurate because of the way variants were identified (i.e., by mapping to a reference genome rather than de novo). As a result, there is a possibility that samples may look more or less like the reference genome in a very systematic way due to biased read mapping. So, measures of diversity, divergence, introgression can be heavily skewed as an artifact of the process, and which haplotype happens to be in the reference genome. The best solution to this issue would be the de novo assembly of the 3 haplotypes in question – bar, checker and *guinea*. Comparing these three would tell you a lot, and could be followed up by using each as a reference for read mapping, although this might not even be necessary. de novo assembly of each may not be practical, in which case more information about the identity of the reference (which haplotype) and coverage stats inside and outside of the focal interval might help indicate whether one is losing data from samples that do not match the reference.

b) An alternative explanation to introgression would be something like a trans-species balanced polymorphism but with one of the two haplotypes lost in *C. guinea*. If selection on the shared haplotype occurred in their common ancestor, this could skew divergence of this haplotype (between species) relative to the other haplotype and much of the genome. The data that make introgression a much more likely explanation are the very different time estimates – 4 million years for speciation vs. 5,000 years divergence. That said, the values presented in Figure 4C (the purple expected bar) and the time estimates presented in the text are derived from a neutral model that doesn't take the possibility of selection into account. For example, the authors could use their genome data to infer a divergence time and historical *Ne* for the two taxa. There are multiple methods that would be useful here, like PSMC, MSMC, G-Phocs, and others. Then, they can test how scenarios such as neutrality, selection / balanced polymorphism, and introgression compare to their empirical data. This approach isn't foolproof, but doing analyses along these lines would provide a more realistic picture of what ancestral variation could produce.

5) As the authors point out in the text, the T-check and checker patterns can be highly variable, which raises the question about the feathers that are shown in Figure 1A (insets). We understand that the specific picture is chosen as a representative, but it would be interesting to get a sense of the variation found within the wing. Are individual feathers in T-check and checker phenotypes highly variable? If so, how can this be explained? Are modifiers also acting locally?

6) Title: consider switching the order of "protein-coding variation" and "introgression of regulatory alleles" in the title. Also consider changing "protein-coding variation", which sounds a bit too general, to something like "missense protein-coding mutation". This would make it to read like: "Introgression of regulatory alleles and a missense protein-coding mutation drive plumage pattern diversity in the rock pigeon".

---

## [Author Response]

Essential revisions:1) There is no linkage/QTL mapping of the color pattern phenotypes. The authors note the prior work on Mendelian genetics, which suggests that there may be useful pedigrees in which alleles at the C locus segregate. The results of association mapping are striking, but there is a step that has been skipped here, which is showing that the C locus phenotypes actually co-segregate with this genomic region. Given the results presented in the study, genetic mapping is not strictly necessary, but this would be a nice addition which should be straightforward.

Thank you for this suggestion. We used an existing cross in our laboratory to test for co-segregation of wing pattern phenotypes and the candidate genomic region identified through genome-wide association mapping. We found that the transmission genetic data, like the genome scans, showed a perfect co-segregation of the candidate genomic region and pigmentation pattern with the expected inheritance pattern (checker alleles are dominant). These results are now reported in the text, the methods are described, and a pedigree showing genotypes and phenotypes is shown in Figure 1—figure supplement 4. To emphasize these points, we now include the following text:

“Pedigree analysis of a laboratory cross also confirmed perfect co-segregation of the checker haplotype and phenotype (Figure 1—figure supplement 4, Supplementary file 1). Thus, a checker haplotype on at least one chromosome appears to be necessary for the dominant melanistic phenotypes, but additional copies of the CNV region are not.”

2) One of the remarkable aspects of color patterning is that there can be such tight spatial control of the mechanisms driving the phenotype. For example, the C allele seems to affect only the wing. Can the authors comment on any possible explanation for this? What happens with NDP expression elsewhere in the body? Is it even expressed? If not, why is it restricted to the wing? What is known about the source of this secreted factor? Incorporating some of these elements into the discussion, even if its speculative, would improve the manuscript.

Different alleles at the *C* locus primarily affect the wing, but subtle effects can sometimes be observed elsewhere. For example, the checker pattern can sometimes extend to the body feathers on the dorsal body, essentially between the wings. Studies in mammals suggest that *NDP* is expressed in other tissues, as we mention in the text, but assaying expression of organs such as the eyes and gonads is highly invasive and requires euthanizing birds. Instead, we assayed expression in feathers in other regions of the body. In order to directly compare our new results to our earlier wing feather results, we assayed allele-specific expression in the same birds that we used for the original wing experiment. We found that *NDP* is expressed in contour (body) and tail feathers, and that checker alleles (1, 2, or 4 copies of the CNV) are expressed significantly higher than bar alleles in both feather types. However, while allele-specific expression trends upward with increasing copy number of the CNV region, these differences are not always significant. These results are now mentioned in the text and shown in Figure 3—figure supplement 3. The main text reads:

“Checker alleles of *NDP* are also more highly expressed in feathers from other body regions (tail and dorsum, Figure 3—figure supplement 3), even though the pigment pattern on these regions is generally similar in bar and checker birds (e.g., both phenotypes have a dark band on the tail). […] Furthermore, because *NDP* expression increases with additional copies of the CNV, the regulatory element probably resides within the CNV itself.”

We do not have direct evidence of the source of *NDP* because the cell populations we harvest for our expression experiments contain several cell types. However, the most likely source of expression in the feather collar is melanocytes. A comment to this effect is now included in the text:

“*NDP* is a short-range signal (Niehrs 2004), so we suspect that this ligand is secreted by melanocytes themselves or by cells in close proximity to them.”

3) The CNV description, subsection “A copy number variant is associated with melanistic wing patterns”, needs to be clarified. Our understanding of the results presented is that while an association exists, there are mismatches – checker and T-check birds that do not have additional copies. So, in terms of the CNV at least, they are like bar? Combined with the absence of the CNV in C. guinea, the results presented argue that the CNV is another associated marker but not causal. Therefore, the current, strong focus on the CNV may be not be warranted and one is left wondering about the possible mechanisms by which the CNV increases NDP expression (see also point #2). Did the authors examine the sequence to look for possible transcription factor binding sites? This may provide insights into why this is a spatially restricted pattern (e.g., a transcription factor that is locally secreted may bind to the CNV).

We understand the confusion surrounding this issue, and we have made changes to the text to help distinguish between the association between the checker haplotype and any degree of the checker phenotype, and the association between the number of copies of the CNV and the amount of plumage pigmentation in checker and T-check pigeons. The most important point is that bar and checker (including T-check) haplotypes in the candidate region are highly differentiated. We now state in the section “A genomic region on Scaffold 68 is associated with wing pattern phenotype”:

“The minimal shared region was defined by haplotype breakpoints in a homozygous checker and a homozygous bar bird, and is highly differentiated from the same region in bar (63.28% mean sequence similarity at informative sites). This region is hereafter referred to as the minimal checker haplotype.”

One copy of a checker haplotype is sufficient to confer the checker phenotype, consistent with the well-characterized dominant inheritance of this trait. Therefore, birds carrying checker alleles with 1 (like *C. guinea*, some checker and T-check pigeons), 2 (checker, T-check), or 4 (checker, T-check) copies of the CNV all have some version of the checker phenotype. We summarize this point in the following way in the section “A copy number variant is associated with variation in melanistic wing patterns” (we added “variation” in the revised subheading for emphasis):

“Consistent with the dominant inheritance pattern of the phenotype, all checker and T-check birds had at least one copy of the checker haplotype. […] Thus, a checker haplotype on at least one chromosome appears to be necessary for the dominant melanistic phenotypes, but additional copies of the CNV region are not.”

The second, separate point is that, among checker and T-check birds, we also observe copy number variation in the part of the candidate region that falls between *EFHC2* and *NDP*. The number of copies of the CNV is significantly associated with the degree of wing plumage pigmentation, as shown in Figure 2C. Copy number is also associated with categorical phenotype (checker, T-check; see Figure 2B), but as we explain in the original text, these categories are not always reliable:

“Checker and T-check birds were grouped together because these two patterns are sometimes difficult to distinguish, even for experienced hobbyists (Figure 1A). Checker birds are typically less pigmented than T-check birds, but genetic modifiers of pattern phenotypes can minimize this difference.”

We now summarize the discussion of the CNV section as follows, which hopefully will provide some additional clarity:

“Together, our analyses show that the checker haplotype is associated with increased pigmentation on the wing shield plumage, resulting in qualitative variation between bar and checker (including T-check) phenotypes. Furthermore, copy number variation is found only in checker haplotypes, and higher numbers of copies are associated with quantitative pigmentation increases in checker and T-check birds only.”

Regarding candidate transcription factor binding sites, we aligned functional elements from the VISTA and REPTILE databases to the CNV region. We report our results as follows:

“To search for known enhancers in the CNV region, we mapped elements from the VISTA (Visel et al. 2007) and REPTILE (He et al. 2017) enhancer datasets to the pigeon genome. […] Further functional work will be required to assess whether this or other sequences in the CNV region act as regulatory elements in *C. livia*.”

4) There are two potential issues with the divergence estimates. a) The estimates may be inaccurate because of the way variants were identified (i.e., by mapping to a reference genome rather than de novo). As a result, there is a possibility that samples may look more or less like the reference genome in a very systematic way due to biased read mapping. So, measures of diversity, divergence, introgression can be heavily skewed as an artifact of the process, and which haplotype happens to be in the reference genome. The best solution to this issue would be the de novo assembly of the 3 haplotypes in question – bar, checker and C. guinea. Comparing these three would tell you a lot, and could be followed up by using each as a reference for read mapping, although this might not even be necessary. de novo assembly of each may not be practical, in which case more information about the identity of the reference (which haplotype) and coverage stats inside and outside of the focal interval might help indicate whether one is losing data from samples that do not match the reference.

To address this potential issue, we performed de novo assemblies, as suggested. We aligned the de novo assemblies to the reference genome to determine regions that overlapped among all three de novo assemblies (bar, checker, *C. guinea*) and called SNPs. The outcome of this analysis was remarkably similar to what we found with our previous short-read mapping approach. We report the outcome of this experiment in the main text and in a new display item, Figure 5—figure supplement 1. Our methods are also described in detail. The new main text section reads as follows:

“The rock pigeon reference genome contains the checker haplotype, which could bias the discovery of SNPs in our resequenced genomes. […] This figure is more recent than our estimate based on more individuals, but the key results are that both estimates are roughly 2 orders of magnitude more recent than the divergence time between species, and the similarity between checker and *C. guinea* sequences is characteristic of within-species rather than between-species variation.”

To avoid too much distraction from the main thread of the paper, details of the results are reported in the figure caption for Figure 5—figure supplement 1. The caption reads as follows:

“Figure 5—figure supplement 1. Expected (purple bar) and observed proportion of shared segregating sites out of 1,462 SNPs in the minimal haplotype region for different pairwise comparisons between de novo genome assemblies from short-read resequencing data for bar, checker, and *C. guinea*. […] Total polymorphic sites between de novo assemblies and the reference (checker) genome: bar-reference, 1346; checker-reference, 12; *C. guinea*-reference, 209.”

b) An alternative explanation to introgression would be something like a trans-species balanced polymorphism but with one of the two haplotypes lost in C. guinea. If selection on the shared haplotype occurred in their common ancestor, this could skew divergence of this haplotype (between species) relative to the other haplotype and much of the genome. The data that make introgression a much more likely explanation are the very different time estimates – 4 million years for speciation vs. 5,000 years divergence. That said, the values presented in Figure 4C (the purple expected bar) and the time estimates presented in the text are derived from a neutral model that doesn't take the possibility of selection into account. For example, the authors could use their genome data to infer a divergence time and historical Ne for the two taxa. There are multiple methods that would be useful here, like PSMC, MSMC, G-Phocs, and others. Then, they can test how scenarios such as neutrality, selection / balanced polymorphism, and introgression compare to their empirical data. This approach isn't foolproof, but doing analyses along these lines would provide a more realistic picture of what ancestral variation could produce.

We thank the reviewers for drawing attention to the fact that the claim of introgression is not supported on the basis of the results in Figure 4C alone. To address this issue, we have expanded and clarified our discussion of the full body of evidence for introgression in the “Signatures of introgression of the checker haplotype” section (some of this text was introduced in our response to comment 4a). We believe this clarification should obviate the need to perform the additional tests for selection. This modified text now reads:

“We measured nucleotide differences among different alleles of the minimal haplotype and compared these counts to polymorphism rates expected to accumulate over the divergence time between *C. livia* and *C. guinea* (Figure 5C, purple bar, see Materials and methods).[…] Thus, humans might have intentionally selected this phenotype, which is linked to life history traits that are advantageous in urban environments, and then built ideal urban habitats for them to thrive (Jerolmack 2008).”

5) As the authors point out in the text, the T-check and checker patterns can be highly variable, which raises the question about the feathers that are shown in Figure 1A (insets). We understand that the specific picture is chosen as a representative, but it would be interesting to get a sense of the variation found within the wing. Are individual feathers in T-check and checker phenotypes highly variable? If so, how can this be explained? Are modifiers also acting locally?

We added a new figure (Figure 1—figure supplement 1) that shows a range of checker phenotypes beyond what is displayed in Figure 1. The new figure includes images of birds in a standing position, with one wing extended, and with individual feathers from the wing shield.

Variation among feathers in an individual bird is indeed an interesting phenomenon. At present, molecular identity of modifiers unknown, and therefore we are unable to determine whether they are expressed locally.

6) Title: consider switching the order of "protein-coding variation" and "introgression of regulatory alleles" in the title. Also consider changing "protein-coding variation", which sounds a bit too general, to something like "missense protein-coding mutation". This would make it to read like: "Introgression of regulatory alleles and a missense protein-coding mutation drive plumage pattern diversity in the rock pigeon".

We agree and have made these changes as suggested.